# Tumor evolutionary directed graphs and the history of chronic lymphocytic leukemia

Jiguang Wang[1,2,3]*[†], Hossein Khiabanian[1,2,3][†], Davide Rossi[4][†], Giulia Fabbri[5,6], Valter Gattei[7], Francesco Forconi[8,9], Luca Laurenti[10], Roberto Marasca[11], Giovanni Del Poeta[12], Robin Foà[13], Laura Pasqualucci[5,6], Gianluca Gaidano[4], Raul Rabadan[1,2,3]*

[1]Department of Biomedical Informatics, Columbia University, New York, United States; [2]Department of Systems Biology, Columbia University, New York, United States; [3]Center for Computational Biology and Bioinformatics, Columbia University, New York, United States; [4]Division of Hematology, Department of Translational Medicine, Amedeo Avogadro University of Eastern Piedmont, Novara, Italy; [5]Institute for Cancer Genetics, Columbia University, New York, United States; [6]Department of Pathology and Cell Biology, Columbia University, New York, United States; [7]Clinical and Experimental Onco-Hematology Unit, Centro di Riferimento Oncologico, Aviano, Italy; [8]Cancer Sciences Unit, Cancer Research UK Centre, University of Southampton, Southampton, United Kingdom; [9]Haematology Department, Southampton University Hospital Trust, Southampton, United Kingdom; [10]Institute of Hematology, Catholic University of the Sacred Heart, Rome, Italy; [11]Division of Hematology, Department of Oncology and Hematology, University of Modena and Reggio Emilia, Modena, Italy; [12]Department of Hematology, Tor Vergata University, Rome, Italy; [13]Department of Cellular Biotechnologies and Hematology, Sapienza University, Rome, Italy

*For correspondence: jw2983@columbia.edu (JW); rr2579@cumc.columbia.edu (RR)

[†]These authors contributed equally to this work

Competing interests: The authors declare that no competing interests exist.

**Abstract** Cancer is a clonal evolutionary process, caused by successive accumulation of genetic alterations providing milestones of tumor initiation, progression, dissemination, and/or resistance to certain therapeutic regimes. To unravel these milestones we propose a framework, tumor evolutionary directed graphs (TEDG), which is able to characterize the history of genetic alterations by integrating longitudinal and cross-sectional genomic data. We applied TEDG to a chronic lymphocytic leukemia (CLL) cohort of 70 patients spanning 12 years and show that: (a) the evolution of CLL follows a time-ordered process represented as a global flow in TEDG that proceeds from initiating events to late events; (b) there are two distinct and mutually exclusive evolutionary paths of CLL evolution; (c) higher fitness clones are present in later stages of the disease, indicating a progressive clonal replacement with more aggressive clones. Our results suggest that TEDG may constitute an effective framework to recapitulate the evolutionary history of tumors.

## Introduction

Cancer is a complex, Darwinian, adaptive clonal evolutionary process, driven by the accumulation of genetic alterations that confer high proliferative and survival advantage (*Merlo et al., 2006*; *Greaves and Maley, 2012*). Recent advances in sequencing technologies have allowed uncovering the most common genetic alterations of many tumors, but the temporal order of most of these alterations is still

**eLife digest** A historical event is often the culmination of the preceding circumstances. The same can be said of cancer as a disease. Cancer results from genetic mutations that disrupt the normal biological processes within a cell, removing the fail-safes that prevent it from growing and reproducing uncontrollably.

Cancer is not caused by just one mutation, and once one gene is malfunctioning, other genes become much more likely to mutate. Although modern sequencing methods have revealed many of the genes that mutate in several different kinds of cancer, uncovering when each of these mutations occurs has been more difficult. Knowing when each mutation occurs could make it easier to predict how the cancer will progress and could also help target cancer treatments more effectively.

Wang, Khiabanian, Rossi et al. have devised a new method of studying the history of genetic mutations of cancer patients. This combines a 'longitudinal' method that looks at how mutations develop in a single tumor by taking samples from it at different times and 'cross-sectional' methods that make predictions based on data collected from a large number of patients. Wang, Khiabanian, Rossi et al. call this method 'tumor evolutionary directed graphs' (TEDG), as it produces a graph that shows how different gene mutations are related to each other. Initial tests showed that the TEDG method could accurately decipher the main chain of events in cancer evolution when used on data collected from at least 30 patients.

Wang, Khiabanian, Rossi et al. then used TEDG on data from 164 tumor samples collected over 12 years from 70 patients with chronic lymphocytic leukemia, the type of leukemia that is most widespread amongst adults in Western countries. This uncovered two separate ways that this cancer may develop, one of which has a higher risk of life-threatening complications.

Knowing which of the two ways chronic lymphocytic leukemia is progressing in a patient could help treat the disease, as each pathway responds differently to different treatments. In addition, understanding the paths that cancer progression follows could also provide early warning signals of the mutations that will occur next. This could help to develop alternative, targeted cancer treatments.

unknown (*Futreal et al., 2004*; *Greenman et al., 2007*; *Santarius et al., 2010*; *Lawrence et al., 2014*). Temporal patterns of genetic alterations may indicate the fate of tumor progression, allowing early diagnosis of tumor subtypes and improving the choice of therapeutic strategies.

To understand the evolutionary history of tumors, several experimental and computational strategies have been used. Longitudinal strategies require samples at multiple time points spanning the clonal tumor evolution process (*Fearon and Vogelstein, 1990*; *Campbell et al., 2010*; *Ding et al., 2010*; *Notta et al., 2011*; *Gerlinger et al., 2012*; *Turajlic et al., 2012*; *Landau et al., 2013*). Landau et al. sampled leukemia cells from 18 CLL patients at two time points, revealing that *SF3B1* and *TP53* mutations are late events in subclonal tumor cells (*Landau et al., 2013*). The study of different stages of colorectal carcinogenesis showed the sequence of genetic events to be *APC*, *KRAS*, and then *TP53* (*Fearon and Vogelstein, 1990*). Another alternative approach is a cross-sectional strategy, which makes use of a large cohort of patients to computationally predict the preferred orders. RESIC is a stochastic process model to identify the order of mutations (*Attolini et al., 2010*), which successfully confirmed the results in colorectal cancer, suggesting that cross-sectional data is informative for the prediction of mutation order. However, RESIC does not consider a critical aspect of carcinogenesis that most tumors are heterogeneous (*Parsons, 2011*).

Following the assumption that different tumors proceed through related temporally ordered alterations, we propose to summarize tumor histories using a newly developed analytical approach that integrates the genomic information from different longitudinally characterized patients. Our method, termed tumor evolutionary directed graphs (TEDG), proceeds in two steps to ensemble in a simplified way cancer clonal evolutionary histories of large number of patients: first, by merging the evolutionary history of each patient, and second, by removing indirect relationships using spectral techniques for network deconvolution (*Feizi et al., 2013*). The resulting TEDG is a directed graph with nodes representing driver genes and arrows representing temporal order of gene lesions. A non-randomly distributed TEDG shows that cancer proceeds in an orchestrated fashion and indicates the main paths and the alternative routes of cancer evolution.

In this study, we have applied TEDG to study the dynamics of the acquisition of alterations in chronic lymphocytic leukemia (CLL), which represents the most common adult leukemia in Western countries (*Hallek et al., 2008*; *Müller-Hermelink et al., 2008*). CLL is an ideal model for studying clonal dynamics because it is possible to collect highly purified sequential samples over time, and its clinical course is well characterized by serial cycles of response, remissions, and relapse ending in some instances with the development of lethal complications such as chemoresistant progression or transformation into an aggressive lymphoma (Richter syndrome) (*Pasqualucci et al., 2011*; *Zenz et al., 2012*; *Fabbri et al., 2013*). No systematic approach has been followed to disentangle and characterize the ensemble of evolutionary histories of this disease. For this purpose, we envision a dual cross-sectional and longitudinal strategy by collecting genomic information from the most common alterations in a cohort of 70 CLL patients spanning over a period of 12 years (2001–2012).

## Results

### Tumor Evolutionary Directed Graphs

To recapitulate and compare the history of genetic alterations in many patients, we propose a framework to infer TEDG by integrating longitudinal and cross-sectional genomic data of cancer patients. First, we reconstruct the sequential network of genetic alterations in each patient by analyzing genomic data from different time points. Specifically, the techniques of high-depth next generation sequencing (NGS) and fluorescence in situ hybridization (FISH) are separately carried out to assess the mutation allele frequency (MAF) and copy number abnormalities (CNA) of selected driver genes. To unify both types of data, and to adjust the MAF of mutations in genes with CNA, we introduce mutation cell frequency (MCF, defined as the fraction of tumor cells with a particular alteration) for quantification of genetic lesions ('Materials and methods', *Figure 1—figure supplement 1*). Based on MCF, we investigate alterations represented in at least 5% of leukemic cells (see examples of CLL patients in *Figure 1—figure supplement 2*). First, if a given genetic lesion is observed to be temporally earlier than another lesion, we connect them with a directed edge to represent their sequential order of development (*Figure 1A*). Second, we pool many sequential networks from different patients to construct an Integrated Sequential Network (ISN). Third, we infer TEDG from ISN by removing indirect associations with spectral techniques and minimal spanning tree algorithm. TEDG is the backbone of ISN, representing an optimal explanation of the mutation order across many patients (*Figure 1B*).

To test TEDG method and also to show how many patients are required to approximate the ground truth, we employ artificial examples by both simulating linear evolution and branching evolution of cancer, where the longitudinal data are generated by one-step Markov process and Nordling's multimutation model (*Nordling, 1953*) (*Figure 2A,B*, 'Materials and methods'). For example, in a cohort of 15 patients with linear evolution, we start the Markov process of each case from no mutations at time zero. Mutation status at three time points 10, 20, and 30 are then successively updated based on the Markov chain transition probability described in 'Materials and methods'. We finally pool all temporal information of those 15 patients to generate ISN (left panel of *Figure 2D*). The TEDG is further deduced by deconvoluting ISN. Suppose $G_{dir}$ is the adjacent matrix of all direct interactions/orders, the observed ISN should be a summary of direct and indirect orders. The deconvolution is formulated by $G_{dir} = G_{obs}(I + \beta G_{obs})^{-1}$, where, $G_{obs}$ is the observed weighted adjacent matrix, $I$ is the identify matrix, and $\beta$ is the scaling factor between zero and one, indicating the degree of deconvolution. To evaluate TEDG, we define the accuracy by how frequently its results match the input model. To calculate the accuracy, we generate 10 simulated datasets of $N$ patients (with three sequential samples per patient). In each simulation, we apply the TEDG analysis to reconstruct the sequential order and compare the result with the input model. We then apply the concept of accuracy to find a reasonable $\beta$ for our simulation, by calculating the accuracy of TEDG with different $\beta \in (0, 1)$ when $N \in \{1, 2, \ldots, 100\}$. *Figure 2C* indicates that the optimal value of $\beta$ is related to the number of samples, but the wide range of high accuracy region suggests that the TEDG results are robust to the parameter selection. With 30 patients, TEDG's accuracy for a linear model is 80%, and for a branching model, it reaches 90% (*Figure 2E–G*, 'Materials and methods').

### Tumor Evolutionary Directed Graph of CLL

In order to investigate the evolutionary history of CLL, we apply the TEDG framework to the driver genetic lesions of this leukemia. We study the most common alterations of 164 temporally sequential

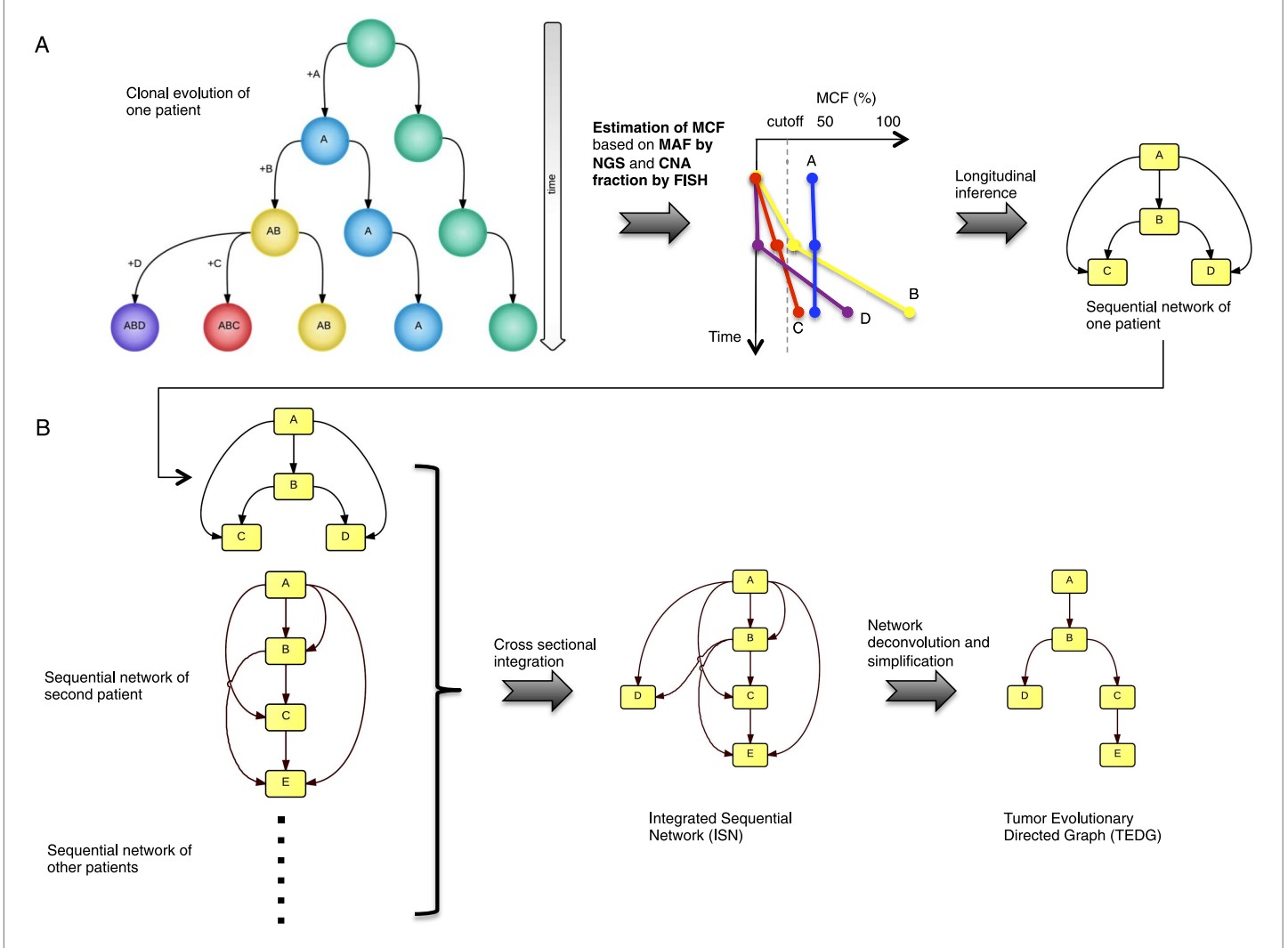

**Figure 1**. Tumor Evolutionary Directed Graph (TEDG) framework. (**A**) A toy example of clonal evolution of one patient. The evolutionary history of four alterations A, B, C, and D is shown in the left panel. We then sample different time points and analyze genomic data. Specifically, for each patient, tagged-amplicon library next generation sequencing (NGS) and fluorescence in situ hybridization (FISH) analyses are carried out at different time points to evaluate the presence and quantify the clonal abundance of possible driver genetic lesions. Then we use mutation cell frequency (MCF) to adjust and unify the data (middle panel). Based on this longitudinal data, we build sequential network of one patient (right panel). CNA: copy number abnormalities. (**B**) Sequential networks derived from different patients (left panel) are further pooled to generate Integrated Sequential Network (ISN), which is a cross-sectional integration of longitudinal data (middle panel). We then infer TEDG by removing the indirect interactions with network deconvolution and simplification algorithms ('Materials and methods'). To construct TEDG, we calculate a minimal spanning tree-based on the deconvolution scores (right panel).

The following figure supplements are available for figure 1:

**Figure supplement 1**. Adjustment of MAF based on copy number data.

**Figure supplement 2**. Relative timing of mutations of 70 CLL patients.

samples from 70 patients by high-depth next generation sequencing (NGS) and fluorescence in situ hybridization (FISH) analysis ('Materials and methods'). Half (35 out of 70) of the patients have at least one subclonal genetic lesion with mutation frequency less than 20% at diagnosis (*Figure 3—figure supplement 1*). We firstly build sequential networks for each patient based on this longitudinal genetic data. Then, we pool all sequential networks to construct the ISN of CLL. We ask whether the genetic

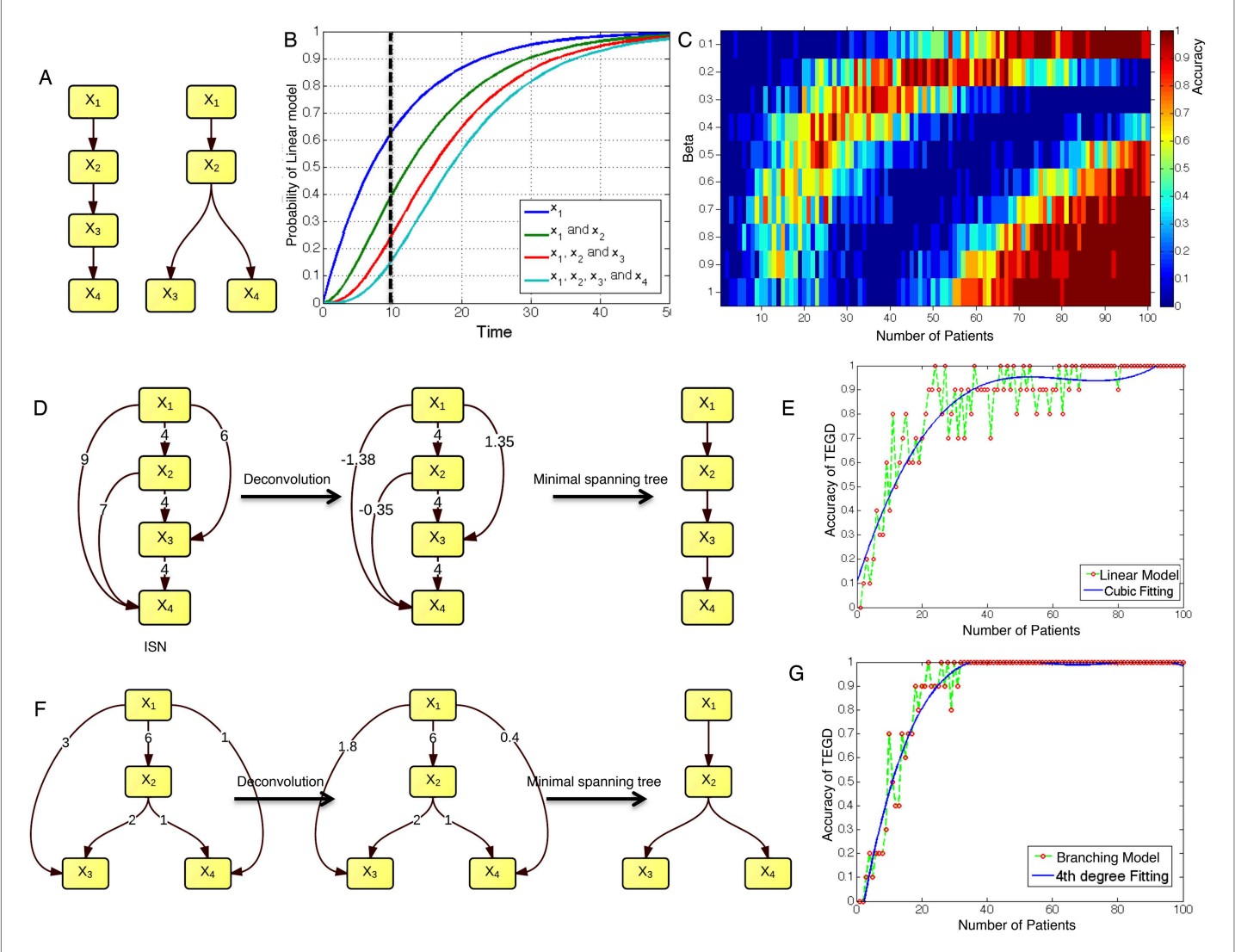

**Figure 2.** The calibration of TEDG framework on the simulation of two basic evolutionary models. (**A**) The representation of linear model and branching model showing the sequential orders of four alterations. (**B**) The probability of observing different mutation profiles by Nordling's multi-mutation model. Specifically, if patient $i$ harbored $k_0$ mutations at time point $t_1$, the probability to observe $k$ more mutations at time point $t_2$ is $\left(1 - e^{-f \cdot (t_2 - t_1)}\right)^k$, where $f$ represents the fitness of the new mutation. (**C**) The selection of parameter $\beta$. The color of heat map represents the accuracy of TEDG analysis, which is defined by the probability of TEDG reconstructing the input model. (**D**) An example showing the deconvolution algorithm on linear model. (**F**) An example showing the deconvolution algorithm on branching model. The left panel represents the ISNs of simulations of 15 patients. The weight of the edges represents the number of patients supporting the corresponding edges. The figures in the middle show the results of network deconvolution, where the numbers on the edge indicate deconvoluted weights. (**E** and **G**) The accuracy of TEDG framework on linear model and branching model at $\beta$ = 0.2.

lesions in CLL are temporally ordered or randomly accumulated and reason that if the genetic alterations driving CLL progression follow a preferential order, there exists a well ordered directed flow in ISN. *Figure 3A* represents a hierarchical layout of ISN, which depicts an ordered structure of lesions in genes represented by sources (nodes with more outgoing arrows) and sinks (nodes with more incoming arrows) (p-value < 0.0001, chi square distribution).

To understand the evolutionary pattern of genetic alterations in CLL, we infer TEDG by removing indirect orders in ISN. The TEDG of CLL (*Figure 3B*) shows a clear pattern of the flow of alterations, revealing that genetic alterations of CLL are sequentially ordered in a branching mode. To statistically assess the temporal pattern of the genetic lesions, we use the binomial test to assess significance by assuming that the number of in-degree and out-degree of each alteration are randomly distributed.

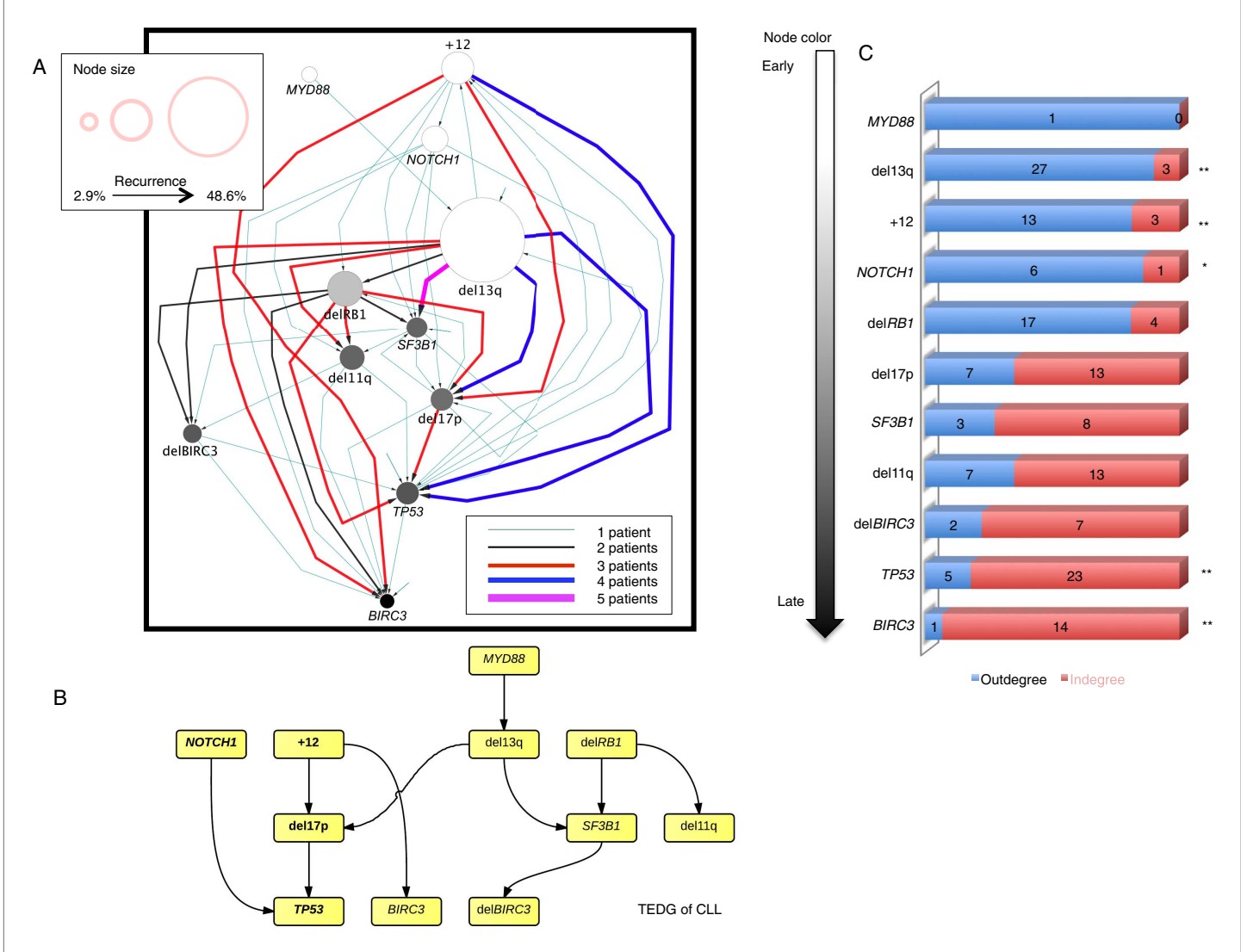

**Figure 3**. Evolutionary network analysis of CLL genetic lesions. (**A**) Network representing the sequential order of genomic alterations in CLL. The nodes in the network represent genetic alterations and the oriented edges (arrows) represent sequential events in different patients where alterations in one gene predate alterations in other genes. The size of the nodes represents the recurrence of alterations in our cohort. The thickness and the color codes of the edges represent the number of patients showing a specific connection between nodes. (**B**) TEDG of CLL, which is the deconvolution of ISN by removing indirect interactions, representing the optimal tree to explain observed orders in CLL patients. (**C**) The order of CLL alterations. We calculate both the sequential in-degree (number of arrows to a node) and out-degree (number of arrows from the node) of each genetic lesion in ISN and use the binomial test to assess the significance by assuming that the number of in-degree and out-degree is randomly distributed. Our null assumption is that if there is no preferential time ordering, there should be the same number of alterations in gene A occurring before alterations in gene B and vice versa, up to statistical fluctuations. Deviations from the null assumption indicate a preferential order in the development of the alterations. All events are sorted by fold change between out-degree and in-degree and all significant early or late events are labeled by *(p-value < 0.05) or **(p-value < 0.01).

The following figure supplements are available for figure 3:

**Figure supplement 1**. Summary of longitudinal data in 70 patients.

**Figure supplement 2**. The order of CLL alterations with and without treatments.

Consistent with TEDG, this analysis suggests the following temporal order of the lesion: mutations of *MYD88*, deletion 13q14, +12, mutations of *NOTCH1*, *RB1* deletion, 17p13 deletion, mutations of *SF3B1*, 11q22-q23 deletion, *BIRC3* deletion, mutations of *TP53*, and mutations of *BIRC3* (*Figure 3C*). Deletion of 13q14, +12 and mutations of *NOTCH1* are significant early events in CLL development,

while mutations of *TP53* and *BIRC3* are significant late events. Though the sample size prevents statistical considerations, *MYD88* mutations might occur even before 13q14 deletion (one of six patients), indicating that activation of the Toll-like receptor pathway could be important in the initiation of a fraction of CLL tumors (*Arvaniti et al., 2011*). The sequential order of the genetic lesions remains consistent (Pearson's correlation = 0.9, p-value < 1e-3) when evolutionary networks are constructed with patients who received chemotherapy (*Figure 3—figure supplement 2*), suggesting that the order of development of the genetic lesions may be affected by therapy (the Pearson's correlation of the order in patients without chemotherapy is 0.4 with p-value > 0.1).

By considering each single type of mutation affecting the same gene as a distinct and independent node, we construct the comprehensive ISN of CLL mutations (*Figure 4A*). It is very difficult to capture useful information directly from ISN, while TEDG simplified the topology by capturing the backbone of tumor evolution (*Figure 4B*). We observe that the monoallelic 13q14 deletion, RB1 deletion, and +12 are significant early events (p-value < 0.01), while the *BIRC3* E537fs and the *TP53* R248Q are significant late lesions (p-value < 0.05) (*Figure 4—figure supplement 1*). Also, the analysis of ISN and TEDG shows that different lesions affecting the same gene may occur in distinct branches and stages. For example, mutations K700E and K666E of *SF3B1* are late events in cases harboring 13q14 deletion, while mutations R273C of *TP53* are late events in cases with +12. Though the sample size prevents statistical considerations, TEDG reveals the H179R and Y234C missense substitutions in *TP53* may be early events, while the R248Q, R273C, P152fs, and N239T substitutions are late events.

Based on pivotal NGS studies, two different evolutionary models have been proposed in CLL, namely gradual linear and branching evolution (*Knight et al., 2012*; *Schuh et al., 2012*). The analysis of our cohort (excluding patients with Richter's transformation, refer to 'Materials and methods' for details) shows that a minority of patients (*n* = 3/60, 5%; FDR = 0.1) are characterized by a significantly decreased or undetectable representation of the founding clone, coupled with a significant increase of a second subclone that represented a small subpopulation at an earlier time point, consistent with a branching evolution model (highlighted by yellow circle in *Figure 5A–B*, and *Table 1*). Interestingly, in all three cases clonal replacement events involve *SF3B1* mutations and occur after treatment (*Figure 5C*), suggesting that the branching evolution model is closely connected to the combination of treatments and the emergence of *SF3B1* mutations (p-value = 0.0016 by Fisher's exact test). This observation implies that, at the time of treatment requirement, limiting the knowledge of disease genetics to the dominant clone will likely be uninformative for accurate therapeutic decisions. Of particular interest in this scenario is the development of therapeutic strategies to prevent the branching evolution of the tumor, with the goal of eradicating dominant as well as minor clones (*Anderson et al., 2011*; *Notta et al., 2011*; *Ding et al., 2012*; *Egan et al., 2012*; *Keats et al., 2012*; *Walker et al., 2012*; *Rossi et al., 2014*).

## Statistical association analysis of TEDG

To further investigate the relationship between driver genetic lesions in TEDG, we assess their associations and anti-associations in a cross-sectional cohort of 1403 CLL patients, of which 1054 cases are informative in at least one lesion (*Figure 6A*, *Table 2*). Most of the co-mutations (connected in red in *Figure 6B*) are experimentally confirmed by previous studies, including: co-occurrence of 17p13 deletion and *TP53* mutations, reflecting a typical two-hit model for tumor suppressor genes; co-occurrence of *BIRC3* deletion, 11q22-q23 deletion, and *BIRC3* mutations; and the relationship between *NOTCH1* mutations and +12 (*Balatti et al., 2012*; *Del Giudice et al., 2012*). In addition to previously reported associated lesions, this large cohort of patients allows the statistical power to reveal other significant and previously unreported co-occurrence interactions, including the co-occurrence of *BIRC3* abnormalities with +12 and *NOTCH1* mutations and the co-occurrence of 13q14 deletion and *BIRC3* deletion. This large cohort also reveals mutually exclusive relationships between +12 and 13q14 deletion and between *NOTCH1* mutations and 13q14 deletion (connected in blue in *Figure 6B*). The statistical association analysis is consistent with the prediction of TEDG and indirectly validates the ability of TEDG in successfully capturing the major information in tumor evolution.

By integrating the topology of TEDG to the association analysis, two distinct and mutually exclusive evolutionary paths of CLL evolution are disclosed. The first evolutionary path involves CLL harboring +12 and *NOTCH1* mutations as early driver events. In this path, clonal evolution proceeds toward the development of *TP53* and *BIRC3* abnormalities. The second evolutionary path involves 13q14 deletion as early driver lesion and proceeds toward the development of *SF3B1* mutations and *BIRC3*

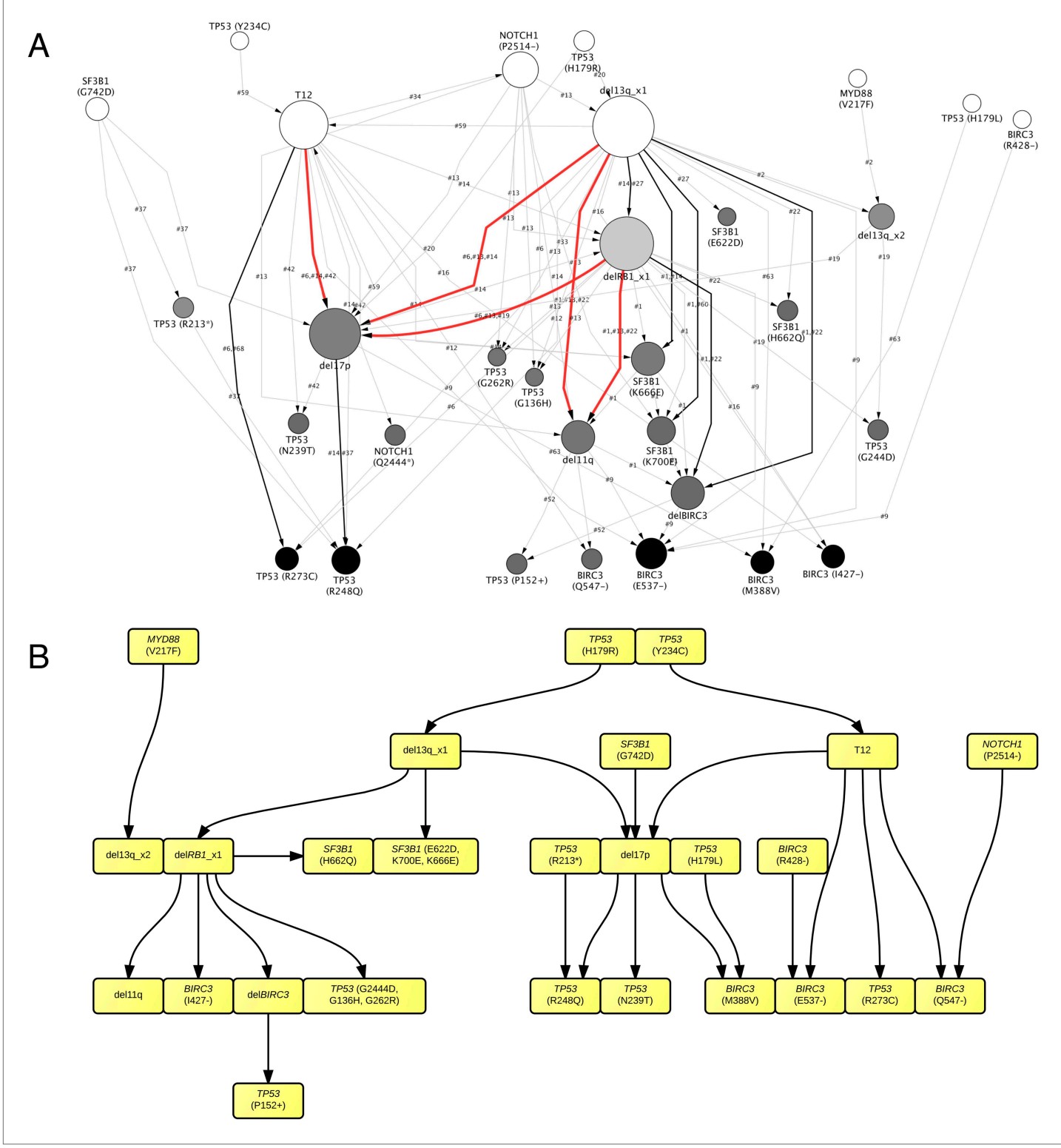

**Figure 4**. TEDG analysis of specific genetic lesions. (**A**) Network representing the sequential order of specific genomic alterations. Two alterations are connected by a directed edge if they are observed to successively appear in at least one patient. The patient ID labels the corresponding edges, which are further colored by the number of patients. Nodes are colored by the fold change between out-degree and in-degree, indicating temporal order of corresponding alterations. The size of the nodes represents the recurrence of the alteration. (**B**) Tumor Evolutionary Directed Graph of 32 specific mutations. Some nodes and arrows in this figure are manually merged or adjusted for the purpose of illustration.

*Figure 4. Continued on next page*

*Figure 4. Continued*

The following figure supplement is available for figure 4:

**Figure supplement 1**. Statistical test of in-degree and out-degree of specific alterations.

abnormalities. Overall, our results are consistent with the different clinico-biological phenotype of +12 CLL and 13q14 deleted CLL and further support the hypothesis that at least two distinct genetic subtypes of CLL exist.

## Rate of allele frequency change

We reason that if a subclone replaces the major clone during tumor evolution, it is because of its higher fitness. Although fitness of a clone is not directly measurable, the growth rate of mutation frequency of drivers in this clone can serve as an indicator. Given this, mutations related to high fitness clones can be identified by extracting the alterations that rapidly change their allele/cell frequency within the tumor. We define the growth rate of each genetic lesion as the average increasing speed of mutation cell frequency per year ('Materials and methods'). Interestingly, the growth rates of late events (17p13 deletion, mutations of *SF3B1*, 11q22-q23 deletion, *BIRC3* deletion, mutations of *TP53*, and mutations of *BIRC3*) are found higher than those of early events (mutations of *MYD88*, 13q14 deletion, +12, mutations of *NOTCH1*, and *RB1* deletion) with p-value = 0.005 by Wilcoxon Rank-Sum Test (*Figure 6C*), revealing that late events may drive higher fitness or more aggressive clones. Note that the initiating lesions are usually clonal, presenting in most of the tumor cells, and therefore do not increase in frequency at the same magnitude as subclonal mutations. However, after eliminating clonal

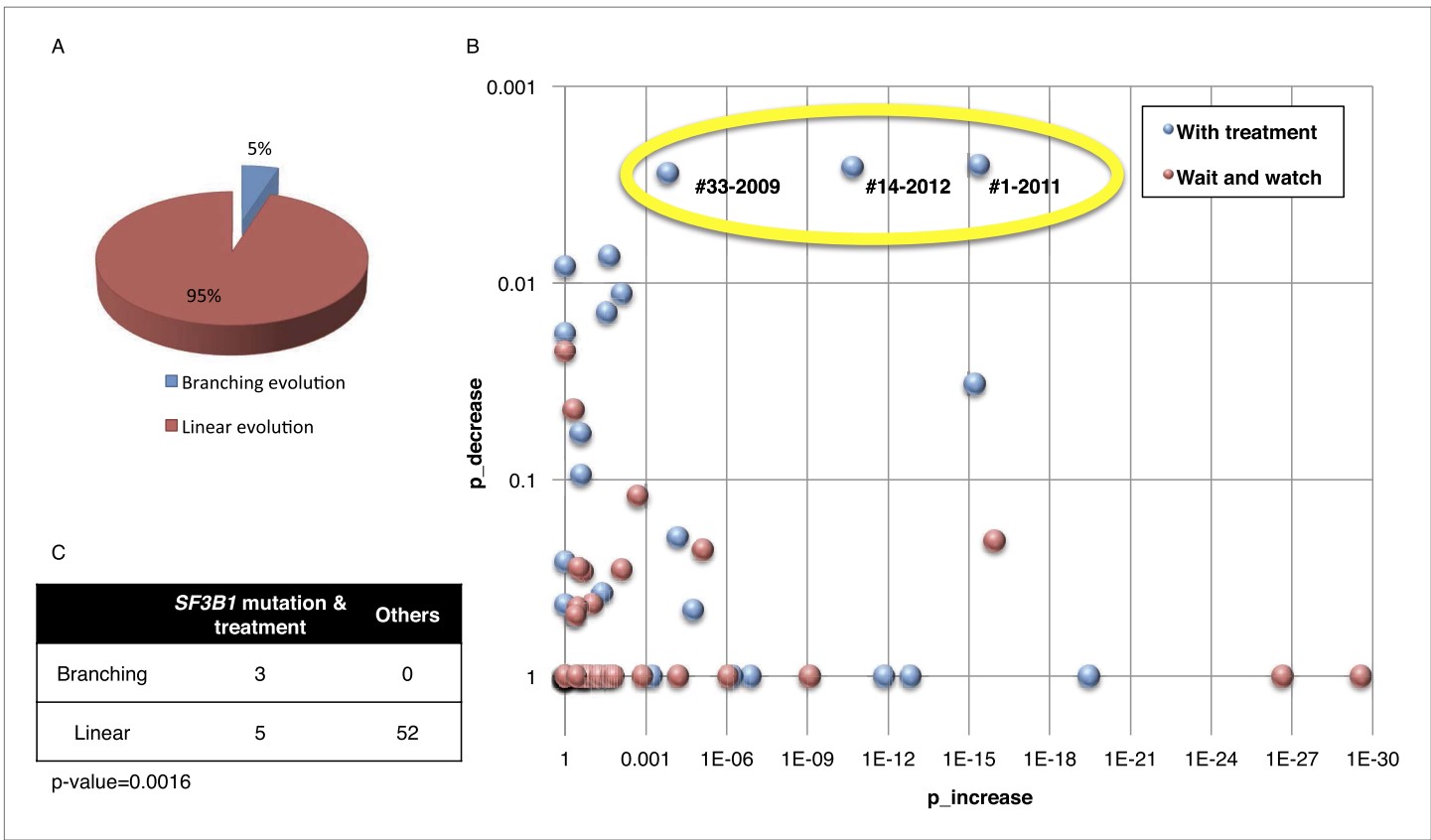

**Figure 5**. Fitting the evolution models. (**A**) Distribution of the clonal evolution pattern in the 60 non-Richter cases. Three of the 60 cases show replacement during tumor progression. (**B**) Scatter plot showing p-values of observing increased and decreased subclones in the 80 samples of the 60 multi-time point patients. Samples with evidence of clonal replacement are located in the right-up corner (highlighted by yellow circle). (**C**) Number of patients following linear vs branched evolution pattern according to *SF3B1* mutational emergence and previous treatment (p-value 0.0016 by Fisher's exact test).

**Table 1.** Patients with branching evolution*

| Patient ID | Sampling time | Treatment | Increased alterations | Decreased alterations | Detectable alterations |
|---|---|---|---|---|---|
| #1 | 2001 | None | – | – | *SF3B1* (K666E), *SF3B1* (K700E), del13q |
| | 2005 | FC | *SF3B1* (K666E) | None | *SF3B1* (K666E), *SF3B1* (K700E), del13q |
| | 2008 | FCR/CAM/RBEN | *SF3B1* (K700E), del11q | None | *SF3B1* (K666E), *SF3B1* (K700E), del13q, del11q |
| | 2011 | BENCAM | *SF3B1* (K700E), del11q, del13q, delBIRC3 | *SF3B1* (K666E) | *SF3B1* (K700E), del13q, del11q, del*BIRC3* |
| #14 | 2004 | RF | – | – | +12, del13q |
| | 2007 | FC/CAM | None | None | +12, del13q |
| | 2010 | BENCAM | *TP53* (R248Q), del17p, *SF3B1* (K666E) | None | +12, del13q, *TP53* (R248Q), del17p, *SF3B1* (K666E) |
| | 2012 | BENDOFA | *TP53* (R248Q), del17p, *SF3B1* (K666E), T12 | del13q | +12, del13q, *TP53* (R248Q), del17p, *SF3B1* (K666E) |
| #33 | 2002 | None | – | – | *NOTCH1* (P2514-) |
| | 2004 | FCR | *SF3B1* (K700E) | None | *NOTCH1* (P2514-), *SF3B1* (K700E) |
| | 2009 | FCR | *SF3B1* (K700E) | *NOTCH1* (P2514-) | *SF3B1* (K700E) |

*FC, fludarabine, cyclophosphamide; FCR, fludarabine, cyclophosphamide, rituximab; CAM, Campath; RBEN, rituximab, bendamustine; BENCAM, bendamustine, Campath; RF, rituximab, fludarabine; BENDOFA, bendamustine, ofatumumab.

genetic lesions with MCF > 20% in the above analysis, we still find the growth rates of late and early events to be significantly different with p-value = 0.0114 by Wilcoxon Rank-Sum Test (***Figure 6— figure supplement 3***).

We define *maximal mutation frequency slope* (MMFS) as the rate of allele frequency change of the fastest-growing clone in a patient (***Table 3***, 'Materials and methods'). MMFS is a function that aims at characterizing the relative fitness of a particular clone carrying a particular mutation. In our cohort, only genetic lesions affecting the *TP53* gene show a statistically significant association with MMFS (p-value = 0.0164, by Wilcoxon Rank-Sum Test). Clinically, the fastest-growing clones are strongly correlated with poor survival and Richter syndrome transformation, consistent with the fact that CLL transformed to Richter syndrome presents significantly higher MMFS values (p-value = 0.002, by Wilcoxon Rank-Sum Test) (***Figure 6—figure supplement 1***). Indeed, by survival analysis, having a clone with high MMFS associates with an approximately threefold significant increase in the hazard of death (HR: 3.17; 95% CI: 1.38–7.40; p-value = 0.005) and a significant shortening of overall survival (47% at 5 years) (***Figure 6—figure supplement 2***). Most deaths (77%) in patients carrying a clone with high MMFS are due to Richter syndrome transformation. Consistently, patients having a clone with high MMFS show an approximately fourfold increased risk of Richter syndrome development (HR: 4.61; 95% CI: 1.54–13.80; p-value = 0.003) and an ~50% of them are projected to develop Richter syndrome at 5 years (***Figure 6—figure supplement 2***).

## Discussion

It is not known whether there is a preferred order of mutations in the development of cancer and how the order of mutations may impact clinical outcomes. We propose a TEDG framework, which is able to integrate longitudinal and cross-sectional genomic data into a directed graph of tumor evolution. The flow in this graph reveals underlying paths of tumor progression. Starting from time-series genomic data in one patient, a sequential network is reconstructed to capture the possible historical events of

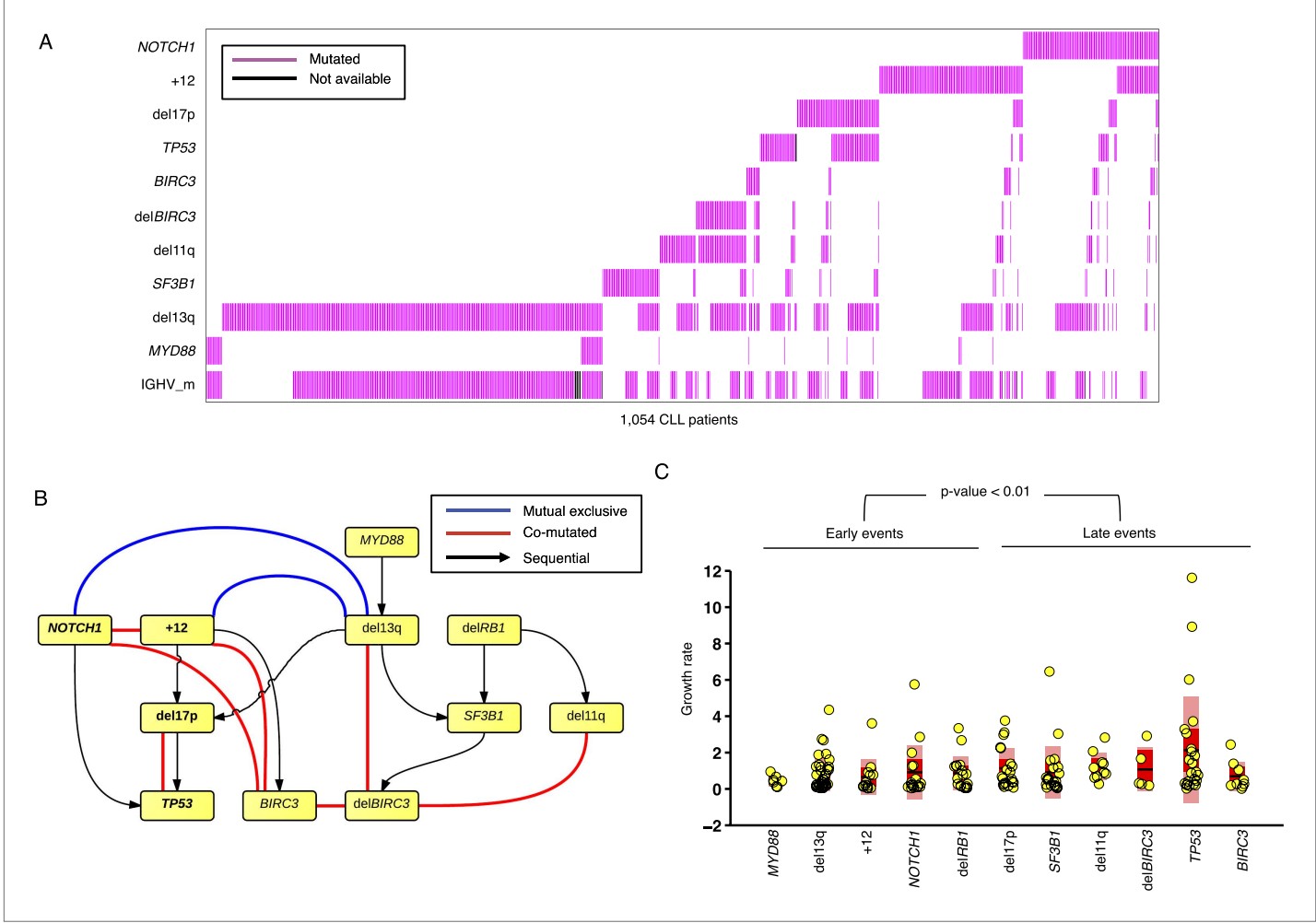

**Figure 6**. Association network of CLL lesions in a larger cohort. (**A**) The mutation status of 1054 CLL patients, which is informative in 11 most common genetic lesions in this leukemia. (**B**) The association analysis of CLL genetic lesions in TEDG. Lesion pairs are connected if they are significantly co-mutated (red) or mutually exclusive (blue). (**C**) Box plot of growth rate of all genetic lesions.

The following figure supplements are available for figure 6:

**Figure supplement 1**. Box plot to show the difference of Richter and non-Richter, and mutations of TP53 in Maximal Mutation Frequency Slope.

**Figure supplement 2**. Survival analysis of fittest genomic alterations.

**Figure supplement 3**. Comparison of the growth rates of subclonal alterations between early and late events.

the evolution in the particular tumor. Integrating sequential data from many patients, we collect the ensemble of tumor histories by ISN, which presents a comprehensive topology of evolutionary landscape. A recent technology of network deconvolution is able to distinguish direct and indirect interactions using spectral methods, which reduce the weights of indirect connections and remove low weighted edges by selecting appropriate cut-offs (**Feizi et al., 2013**). To adapt this method to our particular problem depending on the number of samples, we introduce a degree parameter, β. To gain insights of the selection of beta, we propose a strategy to simulate tumor evolution by a one-step Markov process, with transition probability derived from Nordling's multi-mutation model. A linear and a branching evolution model are separately simulated in a study of four alterations. The proof-of-concept simulation shows that the TEDG strategy can obtain excellent performance in capturing the evolutionary history when the number of cases is beyond 30.

**Table 2.** Clinical features at presentation of the CLL cohorts*

| | TEDG analysis | | | SA analysis | | |
|---|---|---|---|---|---|---|
| | N | Total | % | N | Total | % |
| *IGHV* homology >98% | 37 | 69 | 53.6 | 532 | 1380 | 38.6 |
| del13q14 | 35 | 70 | 50.0 | 682 | 1403 | 48.6 |
| +12 | 19 | 70 | 27.1 | 205 | 1403 | 14.6 |
| del11q22-q23 | 8 | 70 | 11.4 | 136 | 1403 | 9.7 |
| del17p13 | 20 | 70 | 28.6 | 114 | 1403 | 8.1 |
| *TP53* mutation | 22 | 70 | 31.4 | 114 | 1401 | 8.1 |
| *NOTCH1* mutation | 22 | 70 | 31.4 | 150 | 1403 | 10.7 |
| *SF3B1* mutation | 17 | 70 | 24.3 | 100 | 1403 | 7.1 |
| *MYD88* mutation | 6 | 70 | 8.6 | 52 | 1403 | 3.7 |
| *BIRC3* mutation | 10 | 70 | 14.3 | 41 | 1403 | 2.9 |
| *BIRC3* deletion | 6 | 70 | 8.6 | 79 | 1403 | 5.6 |

*TEDG analysis, Tumor Evolutionary Directed Graphs analysis; SA analysis, Statistical Association analysis; *IGHV*, immunoglobulin heavy variable gene; FISH, fluorescence in situ hybridization.

We apply TEDG to CLL and: (i) reconstruct an evolutionary network representing the sequential order of genetic lesions occurring during the course of this disease; (ii) investigate statistical associations and competitiveness of driver genetic lesions to uncover evolutionary paths; and (iii) correlate the order of alterations with the kinetics of changes in their allelic abundance, to identify the molecular alterations associated with the fastest growth of a subclone and their clinical impact on outcome.

Phylogenetic trees are often employed to infer temporal order of gene mutations by assuming that common ancestors are early events, but phylogenetic methods usually require higher number of alterations than the ones available in our cohort. Standard statistical techniques to assess the significance of different branches rely on bootstrapping segregating sites, and the statistical power of these techniques using longitudinal data from cancer patients in few selected driver genes is extremely limited. None of the standard phylogenetic methods such as distance matrix methods, Bayesian phylogenetic methods, or parsimony methods can produce branches with enough bootstrap support (>80%). On the other hand, TEDG works with aggregate data from several patients allowing statistical power for robust estimates.

According to TEDG analysis, the molecular lesions of CLL are temporally ordered in a specific fashion rather than being randomly accumulated. Among recurrent lesions, 13q14 deletion and +12 are initiating events, while mutations of *TP53* and *BIRC3* are late. This observation is consistent with the notion that del13q and +12 occur at a similar prevalence in all CLL phases, including monoclonal B-cell lymphocytosis, a condition that often anticipates overt CLL, thus suggesting that they are early events (*Rawstron et al., 2008*; *Nieto et al., 2009*; *Rossi et al., 2009b*; *Fazi et al., 2011*; *Pasqualucci et al., 2011*; *Kern et al., 2012*). Also, 13q14 deletion has been directly implicated in CLL initiation (*Klein et al., 2010*). On the other hand, late-onset abnormalities are known to accumulate in more advanced phases of the disease, thus suggesting that they are second-hit lesions (*Zenz et al., 2009*; *Rossi et al., 2012*). In our model, mutations of *SF3B1* and 11q22-q23 deletions appear to be acquired at an intermediate time point. This observation is consistent with the clinical observation that mutations of *SF3B1* and 11q22-q23 deletion co-segregate with an intermediate prognostic group of CLL (*Döhner et al., 2000*; *Oscier et al., 2013*; *Rossi et al., 2014*).

By integrating the cross-sectional data of association and anti-association between genetic lesions at CLL diagnosis with longitudinal data of clonal evolution from TEDG, two distinct and mutually exclusive evolutionary paths of evolution emerge. The first evolutionary path involves CLL initially harboring +12 and *NOTCH1* mutations. In this path, clonal evolution proceeds toward the development of *TP53* and *BIRC3* abnormalities. The second evolutionary path involves CLL initially harboring 13q14 deletion and proceeds toward the development of *SF3B1* mutations and *BIRC3* abnormalities. Our data, as well as previous analyses (*Rossi et al., 2013*), show that deletion of 13q14 and +12 are mutually exclusive in CLL. From a clinical standpoint, +12 CLL is known to stand out of typical 13q14 deleted CLL because

**Table 3.** Patients showing high maximal mutation frequency slope (MMFS)

| Patient | MMFS* | Fast growing mutations | MCF1 (%) | MCF2 (%) | dT (moths) | Treatment | Richter syndrome transformation |
|---------|-------|------------------------|----------|----------|------------|-----------|--------------------------------|
| #52 | 117.7 | *TP53* (P152+) | 2.1 | 11.4 | <1 | None | Yes |
| #68 | 9.1 | *TP53* (R273C) | 0.0 | 100.0 | 11 | None | No |
| #51 | 6.7 | *SF3B1* (I704F) | 12.2 | 68.9 | 5 | RCVP | Yes |
| #37 | 6.4 | *TP53* (R248Q) | 3.8 | 90.0 | 10 | RDHAOX | Yes |
| #57 | 5.9 | *NOTCH1* (P2514-) | 0.8 | 94.2 | 12 | None | Yes |
| #47 | 4.6 | del13q | 37.6 | 100.0 | 13 | None | No |
| #14 | 3.9 | del17p | 18.9 | 90.0 | 16 | A | No |
| #42 | 3.4 | *TP53* (N239T) | 0.0 | 100.0 | 16 | RDHAOX | Yes |
| #4 | 3.3 | del17p | 53.6 | 100.0 | 15 | CLB-O | No |
| #54 | 3.0 | *NOTCH1* (P2415-) | 100.0† | 100.0 | 22 | FCO | No |
| #22 | 2.9 | *BIRC3* deletion | 0.0 | 93.9 | 29 | FCR | No |
| #20 | 2.8 | del13q | 0.0 | 100.0 | 36 | CLB | No |
| #63 | 2.5 | *BIRC3* (M388V) | 0.0 | 86.0 | 24 | FCR | Yes |
| #6 | 2.3 | del17p | 0.0 | 92.1 | 34 | CLB | No |
| #38 | 2.3 | *NOTCH1* (P2514-) | 41.7 | 42.9 | 4 | CVP | Yes |
| #13 | 2.2 | *TP53* (G136H) | 0.0 | 100.0 | 45 | RDHAOX | No |
| #1 | 2.1 | del11q | 11.8 | 100.0 | 41 | FCR/A/BR | No |

*MMFS, maximal mutation frequency slope (in standard deviation per year); MCF1, mutation cell frequency of selected mutation at the first time point; MCF2, mutation cell frequency at the second time point; dT, the elapsed time between two samples; RCVP, rituximab, cyclophosphamide, vincristine, prednisone; RDHAOX, rituximab, dexamethasone, high dose cytarabine, oxaliplatin; A, alemtuzumab; CLB-O, chlorambucil, ofatumumab; FCO, fludarabine, cyclophosphamide, ofatumumab; FCR, fludarabine, cyclophosphamide, rituximab; CLB, chlorambucil; CVP, rituximab, cyclophosphamide, vincristine, prednisone; BR, bendamustine, rituximab.

†Total number of the cancer cells with *NOTCH1* alteration does not change, but the allele frequency of the mutation increases because of the deletion of the wild-type allele.

of their atypical cytomorphology and phenotype, the more intense expression of CD20, the preferential nodal presentation and the higher risk of transformation to Richter syndrome. These notions along with our novel observation that +12 and 13q14 deleted CLL proceed through distinct paths of clonal evolutions further support the hypothesis that at least two distinct genetic subtypes of CLL exist.

The presence at diagnosis of fully clonal genetic lesions that are considered late genetic events in CLL evolution, such as *TP53* abnormalities, is already known to have an adverse impact on disease outcome (**Zenz et al., 2008**, **2010**; **Dicker et al., 2009**; **Malcikova et al., 2009**; **Rossi et al., 2009a**; **Gonzalez et al., 2011**). We observe that a fraction of *TP53* abnormalities, though subclonal at presentation, lead to expansion of fitter subclones that progressively predominate with time in the tumor architecture. The clinical impact of this observation is supported by the evidence that harboring higher fitness alterations correlates with poor survival and increased risk of CLL transformation into an aggressive lymphoma, an often-lethal complication known as Richter syndrome.

In this study, we have considered some of the most important and clinically relevant drivers of CLL (**Rossi et al., 2013**). The *ATM* mutations are also commonly found in CLL; however, the *ATM* gene is large and highly polymorphic without well-known hotspots. Therefore, the distinction of its driver mutations from constitutional variants is challenging. Also, from a clinical standpoint, the prognostic relevance of ATM mutations in CLL is still controversial, so we have excluded this gene from the current study (**Lozanski et al., 2012**; **Ouillette et al., 2012**; **Skowronska et al., 2012**).

In conclusion, the application of TEDG to CLL provides the proof-of-principle that this method is able to: (i) improve cancer classification and dissection into genetic subgroups following different paths of clonal evolution; and (ii) anticipate the genetic composition of the progressive/relapsed

disease according to the genetic composition of the tumor clone at the time of diagnosis, including the development of genetic lesions associated with chemorefractoriness. TEDG provides a general framework that could be used to study and compare the evolutionary histories of other tumors.

## Materials and methods

### Samples

We collected 202 CLL patients provided with at least 2 sequential samples and followed for at least 2 years after presentation (median interval between baseline and last sequential sample: 62.8 months, range 24–150 months). 70 out of the 202 patients were informative for *TP53*, *NOTCH1*, *SF3B1*, *MYD88*, or *BIRC3* mutations (which are the most common mutations in this leukemia), overall accounting for 164 paired sequential samples collected at diagnosis, progression, and last follow-up. To analyze the dynamics of those mutations, we carried out high-depth next generation sequencing (NGS) to quantify the mutation allele frequencies at each time point of the disease course and to establish their modifications during leukemia progression. Additionally, copy number abnormalities at 13q14, chromosome 12, 11q22-q23, 17p13, as well as at the *RB1*, and *BIRC3* loci were investigated by FISH. Inclusion criteria for the longitudinal analysis of clonal evolution were: (i) having at least two years of follow-up after diagnosis; and (ii) availability of >2 sequential samples collected at: (a) diagnosis; (b) each progression requiring treatment; (c) last follow-up. CLL diagnosis was according to 2008 IWCLL-NCI criteria and confirmed by a flow cytometry score >3 in all cases. Monoclonal B-cell lymphocytosis (MBL) was excluded. The study was approved by the institutional ethical committee of the Azienda Ospedaliero-Universitaria Maggiore della Carità di Novara affiliated with the Amedeo Avogadro University of Eastern Piedmont, Novara, Italy (Protocol Code 59/CE; Study Number CE 8/11). Patients provided informed consent in accordance with local IRB requirements and Declaration of Helsinki.

CLL samples were extracted from fresh or frozen peripheral blood mononuclear cells (PBMC) isolated by Ficoll-Paque gradient centrifugation. In all cases, the fraction of tumor cells corresponded to 70–98% as assessed by flow cytometry. Matched normal DNA from the same patient was obtained from saliva or from purified granulocytes and confirmed to be tumor-free by PCR of tumor-specific IGHV-D-J rearrangements. High-molecular-weight (HMW) genomic DNA was extracted from tumor and normal samples according to standard procedures. DNA was quantified by the NanoDrop 2000C spectrophotometer (Thermo Scientific, Wilmington, DE). To validate TEDG, 1403 newly diagnosed and previously untreated CLL were enrolled in the study, of whom 931 (66%) were provided with clinical data and regular follow-up (*Table 2*). The frequency of alterations was higher in TEDG analysis group, because only informative patients with known mutated driver genes were included. Cross-sectional investigation of the associations and anti-associations between the most recurrent genetic lesions at diagnosis was based on the entire CLL cohort of 1403 patients. Survival analysis was based on CLL cases provided with clinical data (n = 931). Mutation screening of the *IGHV*, *TP53*, *NOTCH1*, *SF3B1*, *MYD88*, and *BIRC3* genes were performed by Sanger sequencing.

### Estimation of the variant allele frequency

Among cases that were informative for *TP53*, *NOTCH1*, *SF3B1*, *MYD88*, or *BIRC3* mutations at any time point in the disease course, we carried out high-depth NGS to quantify the variant allele frequencies at each stage of their disease. Positions known to harbor *TP53*, *NOTCH1*, *SF3B1*, *MYD88*, or *BIRC3* mutations by Sanger sequencing were amplified from genomic DNA by oligonucleotides containing the gene-specific sequences, along with 10-bp MID tag for multiplexing and amplicon library A and B sequencing adapters. The obtained amplicon library was subjected to deep sequencing on the Genome Sequencer Junior instrument (454 Life Sciences). In order to obtain at least 700-fold coverage per amplicon, no more than 100 amplicons/run were analyzed. The obtained sequencing reads were mapped to reference sequences and analyzed by the Amplicon Variant Analyzer software (Roche) to establish the mutant allele frequency. The sequencing depth in this study is on average 1200×, which is sufficient for a highly sensitive detection of mutations with allele frequency >1% out of the background error noise (*Rossi et al., 2014*).

### Fluorescence in situ hybridization (FISH)

Probes used for FISH analysis were: (i) LSID13S319 (13q14 deletion), CEP12 (trisomy 12), LSIp53 (17p13/*TP53* deletion), and LSIATM (11q2-q23/ATM deletion) (Abbott, Rome, Italy); and (ii) the RP11-177O8 (*BIRC3*) BAC clone. The labeled *BIRC3* BAC probe was tested against normal control

metaphases to verify the specificity of the hybridization. For each probe, at least 400 interphase cells with well-delineated fluorescent spots were examined. Nuclei were counterstained with 4',6'-diamidino-2-phenylindole (DAPI) and antifade reagent, and signals were visualized using an Olympus BX51 microscope (Olympus Italia, Milan, Italy).

## Fluorescence-activated cell sorting (FACS) analysis

The count of CD19+CD5+ cells is a standard assay to define the representation of CLL cells in a diagnostic or research sample as measured by Fluorescence-activated cell sorting (FACS) analysis. A FACSCalibur flow cytometer (Becton–Dickinson) was utilized for the analysis. Expression of CD5 and CD19 was analyzed by combining Peridinin-Chlorophyll-Protein–Cyanine-5.5 (PerCP–Cy5.5)-conjugated anti-CD19 mAbs and fluorescein isothiocyanate-conjugated anti-CD5 mAbs. In order to estimate the proportion of cells co-expressing CD19 and CD5, for each sample events were acquired by gating on low forward and side scatter (FSC/SSC) CD19+ cells, which were further divided into CD5− and CD5+ subsets. Irrelevant isotype-matched antibodies (Becton–Dickinson) were used to determine background fluorescence. FACS data of all samples are listed in *Supplementary file 2*.

## Adjustment of mutation frequency

To consider tumor content, we performed CD19/CD5 FACS analysis to quantify the fraction of tumor cells. To unify FISH and NGS data, and also to adjust MAF of mutations in genes with copy number abnormalities, we introduced mutation cell frequency (MCF). As shown in *Figure 1—figure supplement 1A*, MCF represents the fraction of cancer cells containing particular alterations. Unlike MAF, copy numbers or tumor content do not affect MCF of a point mutation. For instance, if tumor purity is 70% and MAF of a heterogeneous variance in a diploid region is 0.2, then MCF is approximated by 0.2 × 2/0.7 ≈ 0.57. To infer MCF of different types of lesions, we applied the following strategy:

i. MCF of a copy number abnormality (i.e., T12, del11q, del13q_x1, del13q_x2, del17p, delBIRC3, delRB1_x1, delRB1_x2) was calculated by FISH analysis divided by the fraction of CD19/CD5 cells based on FACS, correcting for the tumor purity.

ii. Genes whose copy numbers were not frequently changed, such as *MYD88* and *SF3B1*, were considered as diploid. So the MCF should be close to twofolds of MAF. However, MAF of some mutations might exceed 0.5, because of the random noise introduced in PCR. Here, MCF could be simply defined as a Piecewise function:

$$MCF = \begin{cases} 2 \times MAF, & MAF < 0.5 \\ 1, & MAF \geq 0.5 \end{cases}.$$

But this function was not smooth and it was not able to distinguish MAF = 0.5 and MAF = 0.9. To smooth this function, a Hill function was introduced, assuming

$$MCF = K(MAF)^n \Big/ \left[1 + K(MAF)^n\right],$$

where $K$ and $n$ are parameters. To find optimal parameters, such that $MCF \approx 2 \times MAF$, we optimize the following objective function:

$$\min_{K,n} F(K,n) = \left[ \int_0^{0.5} \left( K(MAF)^n \Big/ \left[1 + K(MAF)^n\right] - 2 \times MAF \right)^2 dMAF \right].$$

A grid-search method was applied to exhaust potential values of $K$ and $n$. Particularly, for $K \in \{2^0, \cdots, 2^9\}$ and $n \in \{1, \cdots, 10\}$, $F$ was calculated by the numerical integration of the above equation. *Figure 1—figure supplement 1B* shows that the optimal solution is $\hat{K} = 16$ and $\hat{n} = 2$. So for mutations of *MYD88* and *SF3B1*,

$$MCF = 16(MAF)^2 \Big/ \left[1 + 16(MAF)^2\right].$$

This Hill function, which was used to smooth the piecewise function, showed a more smooth correlation between MAF and MCF (*Figure 1—figure supplement 1C*). Additionally, by comparing FACS measurement of tumor purity, this Hill function form of MCF is more robust than either MAF without adjustment or simplistic piecewise function in assessing the fraction of cancer nuclei (*Figure 1—figure supplement 1E*).

iii. Genes that are frequently deleted in CLL, such as *TP53* and *BIRC3*, were analyzed in the model without considering copy-neutral LOH or homozygous deletions (*Figure 1—figure supplement 1D*). We assume four different cell types: wild type (upper left), heterozygous deletion (upper right), mutation (lower left), and both mutation and deletion (lower right). The cell fractions were represented by $x_i(i = 1, \cdots, 4)$, respectively. So

$$MAF_{mut} = \frac{x_3 + x_4}{2x_1 + x_2 + 2x_3 + x_4} = \frac{x_3 + x_4}{2 - (x_2 + x_4)} = \frac{MCF_{mut}}{2 - MCF_{del}}$$

and then MCF of mutation depends on MAF of mutation, as well as MCF of the deletion:

$$MCF_{mut} = MAF_{mut}(2 - MCF_{del}).$$

To smooth it, we optimized

$$\min_{K,n} F(K,n; MCF_{del}) = \left[ \int_0^{0.5} \left( K(MAF)^n \Big/ \left[1 + K(MAF)^n\right] - 2 \times MAF + MAF \times MCF_{del} \right)^2 dMAF \right].$$

Here if we use $\hat{K}$ and $\hat{n}$ to represent the optimal solution given $MCF_{del}$, for mutations of *TP53* and *BIRC3*,

$$MCF = \hat{K}(MAF)^{\hat{n}} \Big/ \left[1 + \hat{K}(MAF)^{\hat{n}}\right].$$

The MCF values were then divided by the fraction of CD19$^+$CD5$^+$ cells based on FACS.

## Simulation of linear and branching cancer models

To generate the artificial data for TEDG, we simulate cancer clonal evolution as a one-step Markov process, that is, the transition probability of mutation profile $P\left(\psi_{t_k} | \psi_{t_{k-1}}, \cdots, \psi_{t_1}\right) = P\left(\psi_{t_k} | \psi_{t_{k-1}}\right)$, where $\psi_{tk}$ is the observed mutation profile at time $t_k$, which depended on the mutation profile at time $t_{k-1}$. To define the transition probability, we focused on two simple models of four mutations $x_1, x_2, x_3, x_4$.

In the linear evolution model, we assumed that all mutations were mutated in a linear order shown in the left panel of *Figure 2A*. All possibilities of mutation status are

$$\left\{ \pi_0 = \varphi, \pi_1 = (x_1), \pi_2 = (x_1, x_2), \pi_3 = (x_1, x_2, x_3), \pi_4 = (x_1, x_2, x_3, x_4) \right\}.$$

According to Nordling's multi-mutation model (*Nordling, 1953*), the transition probability was defined as

$$P\left(\psi_{t_k} = \pi_j | \psi_{t_{k-1}} = \pi_i\right) = \begin{cases} \left[1 - e^{-f \cdot (t_k - t_{k-1})}\right]^{j-i} - \left[1 - e^{-f(t_k - t_{k-1})}\right]^{j-i+1}, & j > i; \\ e^{-f \cdot (t_k - t_{k-1})} & , & j = i; \\ 0 & , & j < i. \end{cases}$$

where $f$ represents fitness of the new mutation. The transition probability is shown in *Figure 2B*. The longitudinal data were then generated for each patient following the above model. To simplify the model, we fixed the number of time points (three) and the length of interval between time points (ten). We asked whether or not TEDG framework can reconstruct the order of mutations and how many patients are required. To answer this question, we applied a grid-search strategy to optimize

parameter $\beta$ and number of patients. Specifically, all possible combinations of parameter $\beta$ from 0 to 1 (step by 0.1) and patient number from 1 to 100 (step by 1) are exhausted. For each combination, we randomly simulated the cancer patients and applied TEDG framework to reconstruct the order for 10 times. The frequency of reconstructing the exact order was defined as the accuracy of TEDG. **Figure 2D** shows an example with $\beta$ equals 0.2 and the number of patients equals 15. Edge weights of ISN present the number of simulated patients with one mutation happening before another. The techniques of deconvolution and minimal spanning tree reconstruct the real structure by removing indirect interactions. **Figure 2E** summarizes the correlation between number of patients and optimal accuracy of TEDG.

Different from linear model, the branching model assumes that $x_3$ and $x_4$ are independently following the mutation of $x_2$. All possibilities of mutation status are

$$\left\{ \pi_0 = \varphi, \pi_1 = \left(x_1\right), \pi_2 = \left(x_1, x_2\right), \pi_3 = \left(x_1, x_2, x_3\right), \pi_4 = \left(x_1, x_2, x_4\right) \right\}.$$

According to Nordling's multi-mutation model (**Nordling, 1953**), when $j \leq 2$, the transition probability is the same as linear model:

$$P\left(\psi_{t_k} = \pi_j | \psi_{t_{k-1}} = \pi_i\right) = \begin{cases} \left[1 - e^{-f \cdot (t_k - t_{k-1})}\right]^{j-i} - \left[1 - e^{-f \cdot (t_k - t_{k-1})}\right]^{j-i+1}, & j > i; \\ e^{-f \cdot (t_k - t_{k-1})} & , & j = i; \\ 0 & , & j < i. \end{cases}$$

When $j > 2$, the transition probability is

$$P\left(\psi_{t_k} = \pi_j | \psi_{t_{k-1}} = \pi_i\right) = \begin{cases} \left[1 - e^{-f \cdot (t_k - t_{k-1})}\right]^{3-i}, & i < 3; \\ 1 & , & i = j; \\ 0 & , & otherwise. \end{cases}$$

**Figure 2F** shows an example of branching model with $\beta$ equals 0.2 and number of patient equals 15. **Figure 2G** summarized the correlation between number of patients and optimal accuracy of TEDG in branching model.

## Network construction and analysis

To construct the sequential network of alterations for each tumor, we monitored the presence or absence of each genetic lesion in each sample. By collecting the clonal representation of each lesion from NGS and FISH analysis, we defined the status of each lesion with a cut-off of 5%. If the frequency of a genetic lesion is larger than 5%, we will name it as present; otherwise absent. If event A predates event B, we added a directed link between A and B. The ISN was constructed by pooling all sequential networks from different patients. To simplify ISN, we removed self-loops by subtracting the weight of weaker direction. To hierarchically layout the simplified ISN, we use yFiles Hierarchical Layout algorithm in Cytoscape 2.8.2, which well represents main direction or 'flow' in a directed network. With this method, nodes were placed in hierarchically arranged layers and the nodes within each layer are ordered in such a way that minimizes the number of edge crossings (**Smoot et al., 2011**). To statistically test whether the alterations are temporally ordered or randomly accumulated, we assume that in-degree (the number of incoming arrows) $d_{in}^i$ is equal to out-degree (the number of outgoing arrows) $d_{out}^i$ for each node $i$, and then $\sum_i \frac{\left(d_{in}^i - d_{out}^i\right)^2}{d_{out}^i}$ follows a chi square distribution with degree of freedom $n - 1$, where $n$ is the number nodes in the network. To test the statistical association of lesions in TEDG, we counted the number of samples carrying each pair of lesions and then calculated the p-value based on hypergeometric distribution to test whether the two genetic lesions are independent or not. To consider the effect of multi-hypothesizes, we corrected p-values with Bonferroni method and made a cut-off of 0.05. If two lesions were significantly co-mutated, a red link was added, while if they were significantly mutually exclusive, a blue edge was added.

Suppose $G_{dir}$ is the adjacent matrix of all direct interactions/orders, the simplified ISN should be a summary of direct and indirect orders in the deconvolution formula $G_{dir} = G_{obs}(I + \beta G_{obs})^{-1}$, where,

$G_{obs}$ is the observed weighted adjacent matrix, $I$ is the identify matrix, and $\beta$ is a scaling factor between zero and one indicating the degree of deconvolution. The resulting matrix of deconvolution formula indicates the score of an edge to be a direct interaction (edge weights in the middle panel of **Figure 2D,F**). We then introduced a minimal spanning tree-based method to determine the final TEDG, in which we transformed the deconvoluted weights with a negative exponential function, and then calculated the minimal spanning tree with Prim's algorithm.

### Fitting the evolutionary models

We developed Fit the Evolutionary Model (FEM) to properly fit the evolutionary model by systematically identifying clonal replacement in all sample pairs. FEM defines $Z$-scores for variant allele frequency by NGS and percentage of nuclei harboring abnormalities by FISH to represent normalized changes of the frequency of the lesions. For $i$th mutation in $j$th patient at the $k$th time point $t_k$

$$Z_i^j(t_k) = \frac{f_i^j(t_k) - f_i^j(t_{k-1})}{\sigma_{seq}\sqrt{\frac{f_i^j(t_k) + f_i^j(t_{k-1})}{2}\left(1 - \frac{f_i^j(t_k) + f_i^j(t_{k-1})}{2}\right)}},$$

where $f$ is the mutation frequency of some particular gene mutation and $\sigma_{seq}$ is its standard variation. Similarly, for the $i$th copy number change in the $j$th patient at the $k$th time point $t_k$

$$Z_i^j(t_k) = \frac{f_i^j(t_k) - f_i^j(t_{k-1})}{\sigma_{FISH}\sqrt{\frac{f_i^j(t_k) + f_i^j(t_{k-1})}{2}\left(1 - \frac{f_i^j(t_k) + f_i^j(t_{k-1})}{2}\right)}},$$

where $f$ is the frequency nuclei harboring the abnormality of some particular gene mutation and $\sigma_{FISH}$ is its standard variation.

The change of genetic lesion frequency is a synergic effect of treatment, tumor progression, and experimental noises, such as sequencing error or change of tumor purity. To obtain the variance caused by background noises, FEM robustly fit $\sigma_{seq}$ and $\sigma_{FISH}$ by eliminating data obviously affected by treatments or tumor progression. Based on this, we could calculate p-values of each genetic lesion in all sample pairs. To assess whether there was a significantly increased (or decreased) subclone in a given sample pair, we use Fisher's combinational test to combine p-values of all increased (or decreased) genetic lesions. The resulting p-values were separately defined as P_increase and P_decrease. Sample pairs that were significant in both P_increase and P_decrease were defined as sample pairs containing replacement events, which were further fit to the branching evolution model. All the others were compatible with the gradual linear model.

### Growth rate and maximal mutation frequency slope (MMFS)

To investigate the fitness of genetic lesions, we defined growth rate and the maximal mutation frequency slope (MMFS). For the $i$th mutation in the $j$th patient at the $k$th time point $t_k$, mutation frequency slope is defined as $s_i^j(t_k) = \frac{z_i^j(t_k)}{t_k - t_{k-1}}$, where $z$ is the $Z$-scores defined above. The growth rate of $i$th mutation in the $j$th patient at the $k$th time point $t_k$ is defined as $g_i^j(t_k) = max\{s_i^j(t_k), 0\}$. Furthermore, for the $j$th patient, MMFS is defined as $s^j = max_i\{max_k[s_i^j(t_k)]\}$.

### Survival analysis

Overall survival (OS) was measured from date of initial presentation to date of death from any cause (event) or last follow-up (censoring). The cumulative probability of Richter syndrome transformation was measured from date of initial presentation to date of the biopsy documenting Richter syndrome transformation (event), death or last follow-up (censoring). Survival analysis was performed by the Kaplan–Meier method. The crude association between time-fixed exposure variables at diagnosis and survival was estimated by Cox proportional hazard regression. The analysis was performed with the Statistical Package for the Social Sciences (SPSS) software v.20.0 (Chicago, IL).

## Candidate gene mutation screening

The mutation hotspots of *TP53* (exons 4–9, including splicing sites; RefSeq NM_000546.5), *NOTCH1* (exon 34, including splicing sites; RefSeq NM_017617.2), *SF3B1* (exons 14, 15, 16, 18, including splice sites; RefSeq NM_012433.2), *MYD88* (exons 3, 5, including splicing sites; RefSeq NM_002468.4), and *BIRC3* (exons 6–9, including splicing sites; RefSeq NM_001165.4) genes were analyzed by PCR amplification and DNA direct sequencing of high-molecular weight genomic DNA. Sequences for all annotated exons and flanking splice sites were retrieved from the UCSC Human Genome database using the corresponding mRNA accession number as a reference. PCR primers, located ~50 bp upstream or downstream to target exon boundaries, were either derived from previously published studies or designed in the Primer 3 program (http://frodo.wi.mit.edu/primer3/) and filtered using UCSC in silico PCR to exclude pairs yielding more than a single product. All PCR primers and conditions are listed in *Supplementary file 1*. Purified amplicons were subjected to conventional DNA Sanger sequencing using the ABI PRISM 3100 Genetic Analyzer (Applied Biosystems) and compared to the corresponding germline sequences using the Mutation Surveyor Version 4.0.5 software package (SoftGenetics) after automated and/or manual curation. Of the evaluated sequences, 99% had a Phred score of 20 or more and 97% had a score of 30 or more. Candidate variants were confirmed from both strands on independent PCR products. The following databases were used to exclude known germline variants: Human dbSNP Database at NCBI (Build 136) (http://www.ncbi.nlm.nih.gov/snp); Ensembl Database (http://www.ensembl.org/index.html); The 1000 Genomes Project (http://www.1000genomes.org/); five single-genome projects available at the UCSC Genome Bioinformatics resource (http://genome.ucsc.edu/). Synonymous variants, previously reported germline polymorphisms and changes present in the matched normal DNA were removed from the analysis.

## *IGHV-IGHD-IGHJ* rearrangement analysis

PCR amplification of *IGHV-IGHD-IGHJ* rearrangements was performed on high molecular weight genomic DNA using *IGHV* leader primers or consensus primers for the *IGHV* FR1 along with appropriate IGHJ genes, as previously described. PCR products were directly sequenced with the ABI PRISM BigDye Terminator v1.1 Ready Reaction Cycle Sequencing kit (Applied Biosystems) using the ABI PRISM 3100 Genetic Analyzer (Applied Biosystems). Sequences were analyzed using the IMGT databases and the IMGT/V-QUEST tool (version 3.2.17, Université Montpellier 2, CNRS, LIGM, Montpellier, France).

## Acknowledgements

We thank Dr Riccardo Dalla-Favera, Alexandra Jacunski, Dr Joseph Chan, and Dr Yong Wang for their helpful comments. LP is on leave from the University of Perugia Medical School.

## Additional information

### Funding

| Funder | Grant reference number | Author |
| --- | --- | --- |
| National Center for Advancing Translational Sciences | UL1 TR000040 | Jiguang Wang |
| National Institutes of Health | 1 U54 CA121852-05 | Hossein Khiabanian, Raul Rabadan |
| National Institutes of Health | 1R01CA185486-01 | Jiguang Wang, Raul Rabadan |
| National Institutes of Health | 1R01CA179044-01A1 | Jiguang Wang, Raul Rabadan |
| Associazione Italiana per la Ricerca sul Cancro | Special Program Molecular Clinical Oncology, 5x1000, No. 10007 | Robin Foà, Gianluca Gaidano |
| Associazione Italiana per la Ricerca sul Cancro | My First AIRC Grant 2012 | Davide Rossi |

| Funder | Grant reference number | Author |
|---|---|---|
| Ministero dell'Istruzione, dell'Università e della Ricerca | Futuro in Ricerca 2002 and 2012 | Davide Rossi |
| Istituto Nazionale di Fisica Nucleare | PRIN 2008 | Gianluca Gaidano |
| Istituto Nazionale di Fisica Nucleare | PRIN 2009 | Davide Rossi |
| Ministero della Salute | 2008 and 2010 | Davide Rossi |
| Alexander and Margaret Stewart Trust | | Raul Rabadan |

The funders had no role in study design, data collection and interpretation, or the decision to submit the work for publication.

## Author contributions

JW, Performed statistical analysis, Conception and design, Analysis and interpretation of data, Drafting or revising the article; HK, Performed statistical analysis, Analysis and interpretation of data, Drafting or revising the article; DR, Performed statistical analysis, Acquisition of data, Drafting or revising the article; GF, LP, Analysis and interpretation of data, Drafting or revising the article; VG, Collected patients and samples, Acquisition of data, Analysis and interpretation of data; FF, LL, RM, GDP, RF, Acquisition of data, Analysis and interpretation of data; GG, Acquisition of data, Analysis and interpretation of data, Drafting or revising the article; RR, Conception and design, Analysis and interpretation of data, Drafting or revising the article

## Ethics

Human subjects: The study was approved by the institutional ethical committee of the Azienda Ospedaliero-Universiataria Maggiore della Carita di Novara affiliated with the Amedeo Avogadro University of Eastern Piedmont, Novara, Italy (Protocol Code 59/CE; Study Number CE 8/11). Patients provided informed consent in accordance with local IRB requirements and Declaration of Helsinki.

## Additional files

### Supplementary files

• Supplementary file 1. PCR primers and conditions.

• Supplementary file 2. FACS Data.

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
