## [Decision Letter]

Thank you for sending your work entitled “Tumor Evolutionary Directed Graphs and the History of Chronic Lymphocytic Leukemia” for consideration at *eLife*. Your article has been favorably evaluated by Chris Ponting (Senior editor), a Reviewing editor, and 2 reviewers. The Reviewing editor has assembled the following comments to help you prepare a revised submission.

Overall, the reviewers found the work interesting but raised a number of concerns that need to be addressed regarding the analysis of the data as well as clarifications on the design and presentation of the work.

Major comments:

1) Relative timing of mutations: It is not entirely clear in the Materials and methods or main text exactly how the temporal order of mutations is inferred. At one point, mutations are described as 'present' or 'absent' on the basis of allele fraction >5% and compared across time points. Really, the only true method for inferring temporal ordering is through reconstruction of phylogenetic trees; this can be difficult from bulk sequencing data, but there are methods emerging to do this.

2) Copy-number adjusted allele fractions: On a similar note, it is not clear that the fraction of cells carrying a point mutation has been adjusted for the copy number at that locus. This is especially important for the *TP53* point mutations when the other allele has undergone LOH. Also, how are double hits in a given gene dealt with in the graph method?

3) Rate of increase: The argument that later mutations show more rapid increase than earlier mutations is difficult to sustain on the basis of these data. The initiating lesion, by definition, will be present in all the tumor cells, and therefore cannot increase at the same magnitude as later mutations. Given that it is a prerequisite for entry into the study that the patient be diagnosed with CLL, one would imagine that, at that diagnostic time point, all the earliest mutations might already be clonal, and therefore will not change over time.

4) A question for another type of inference approach is: How do we know ground truth? The only way to know is if there were a set of samples from which there was frequent sampling, and if there were within clinical defined groups. Robustness of the network is dependent on how good the data is going in. While they do mention that they selected 70 samples for TEDG for which there were at least two sequential samples, in all likelihood, 2 samples are inadequate for defining the hierarchical relationships. There needs to be some table that provides information on the distribution of sequential samples per patient, and what types of time intervals between sequential samples. While a figure about the samples is provided as Figure 3—figure supplement 1, this is very hard to digest. This is a very very heterogeneous dataset, with 10 of 70 proceeding to Richters, more than half with treatment. A question is whether or not there are enough samples, by the time that one breaks down the patients into discrete clinical groups, to understand the hierarchical relationships.

Related to this, the numbers of samples are very confusing. This is what the reviewer understood: it seems that they start out with 202 patients, but only 70 had known highly recurrent drivers. 10 of the 70 had Richters Transformation, and how many have been excluded? By the time we get to Figure 4, there are 32 samples, but there is no discussion how this number came about. In general, in what percentage of sample was a polyclonal tumor population or a subclonal driver identified at the time of diagnosis?

[Editors' note: further revisions were requested prior to acceptance, as described below.]

Thank you for resubmitting your work entitled “Tumor Evolutionary Directed Graphs and the History of Chronic Lymphocytic Leukemia” for further consideration at *eLife*. Your revised article has been favorably evaluated by Chris Ponting (Senior editor), a member of the Board of Reviewing Editors, and the original two reviewers. The manuscript has been improved but there are two remaining major issues that will need to be addressed before acceptance, as outlined below. We have copied the exact wording of the reviewer to facilitate the response.

Major issues:

1) A major revision has been the addition of the concept of MCF. This was done to address the concern of adjusting for local copy number changes that could affect estimation of the clone size with a gene alteration. Unfortunately, the approach taken by the authors to calculate MCF seems not well defined. This reviewer is concerned that sampling error in VAF values might fluctuate above or below 0.5 in ways that have a large influence on the inferred MCF. No convincing data is presented by the authors to justify their novel and elaborate seeming approach. Several approaches for correcting mutation VAF values for tumor purity and somatic copy number changes have been described (e.g. Carter et al., 2012, Nat Biotech; [26], Cell; Fischer et al., 2014, Cell Reports). Notably, these methods rely on analysis of germline heterozygous SNPs at the mutant locus to infer CN-LOH, which is a much more robust approach. Could the authors simply use one of these established (and principled) approaches? This would leverage their exome data more fully.

2) They rely on FISH analysis for estimations of trisomy 12 and deletions of various chromosomal regions. This is not accurate, as it is not possible to correct for purity when using FISH data. In addition, copy neutral LOH cannot be inferred from FISH.

---

## [Author Response]

We found the comments extremely helpful, and according to the suggestions we have made several major revisions summarized as follows:

A) We introduce the concept of mutation cell frequency (MCF) to adjust mutation allele fractions using copy-number information. The calculation of MCF not only eliminates potential bias caused by “two-hit” alterations but also unify next generation sequencing (NGS) and fluorescence in situ hybridization (FISH) for data integration. After the adjustment, all our main results are consistent as shown in the following.

B) To address the major comment #1, we introduce a phylogenetic tree method to infer timing order of mutations. Our analysis shows that the phylogenetic tree method and TEDGs differ in capturing the effect of competition between different subclones. However, the main results inferred from both methods are consistent, indicating the robustness of the overall analysis (see detailed response below).

C) To address the concern that the growth rates of initial clonal events are biased, we only consider the increase rates of early subclonal events (mutation frequency <20% at diagnosis). In the new analysis this conclusion remains.

D) To clarify the confusion of the sample size in our study, we generate a new figure with a clear statement of how we select patients, and how many samples of each patient. To show how many patients are necessary to reveal the ground truth, we simulate the evolutionary and sampling process by an artificial example. The results indicate 30 patients will be enough to reconstruct a simple evolutionary graph.

All comments are addressed in detail as follows:

*2) Copy-number adjusted allele fractions: on a similar note, it is not clear that the fraction of cells carrying a point mutation has been adjusted for the copy number at that locus. This is especially important for the* TP53 *point mutations when the other allele has undergone LOH. Also, how are double hits in a given gene dealt with in the graph method?* (Note: comment #2 is addressed first because other comments will depend on the adjustment of mutation data.)Author response image 1.Definition of mutation cell frequency.MAF: mutation allele frequency; MCF: mutation cell frequency. The black lines within the circles represent DNA copies, and the crosses represent mutations. The contingent table shows the difference between MAF and MCF.

This is a good suggestion. To consider potential bias caused by copy number abnormalities, we introduce mutation cell frequency (MCF) instead of Mutation Allele Frequency (MAF). As shown in Figure 7, MCF represents the fraction of cells containing alterations. Different from MAF, MCF of a mutation is not affected by copy numbers. To infer MCF of different types of alterations, we apply the following strategy:

i) The percentages of cells harboring the copy number abnormality (i.e. +12, del11q, del13q_x1, del13q_x2, del17p, delBIRC3, delRB1_x1, delRB1_x2) are estimated by FISH analysis.Author response image 2.Optimization of Hill function by grid search method.*z*-axis indicates the objective function, *x*-axis and *y*-axis are parameters of Hill function.

ii) Genes that are not frequently deleted or amplified in CLL, such as *NOTCH1*, *MYD88*, and *SF3B1* are considered as diploid. So the MCF should be close to two folds of MAF. However, MAF of some mutations may exceed 0.5, because of the random noise introduced in PCR process or because sometimes the diploid assumption does not hold. MCF could be defined in a simple Piecewise function:MCF={2×MAF,MAF<0.51,MAF≥0.5.

But this function is not smooth and it is not able to distinguish MAF=0.5 and MAF=0.9. To smooth this function, a Hill function strategy is introduced.MCF=K(MAF)n[1+K(MAF)n],

Assume where *K* and *n* are parameters of the Hill function. To find optimal *K* and *n* such that MCF≈2×MAF, we optimize the following objective function:minK,nF(K,n)=[∫00.5(K(MAF)n[1+K(MAF)n]−2×MAF)2dMAF].Author response image 3.Optimal correlation between MAF and MCF.

A grid search method is applied to exhaust potential values of *K* and *n*. Particularly, for K∈{20,⋯,29}, and n∈{1,⋯,10},
*F* is calculated by the numerical integration of the above equation. Figure 8 shows the optimal solution is K^=16 and n^=2. So for mutations of *NOTCH1*, *MYD88*, and *SF3B1:*MCF=16(MAF)2[1+16(MAF)2].

This Hill function, which is used to smooth the piecewise function, shows a more reasonable correlation between MAF and MCF (Figure 9).

iii) Genes that are frequently deleted in CLL, such as *TP53* and *BIRC3*, are analyzed in the model without considering copy-neutral LOH and homozygous deletions (Figure 10). We assume four different genotypes: wild type (upper left), heterozygous deletion (upper right), mutation (lower left), and both mutation and deletion (lower right). The cell fractions are represented by xi(i=1,⋯,4), respectively. SoMAFmut=x3+x42x1+x2+2x3+x4=x3+x42−(x2+x4)=MCFmut2−MCFdel

and then MCF of mutation depends on MAF of mutation, as well as MCF of the deletion:MCFmut=MAFmut(2−MCFdel).Author response image 4.Genotypes of two-hit model.Black lines within the circles represent copies of DNA, and crosses on the lines indicates point mutations. *χ*_*1*_, *χ*_*2*_, *χ*_*3*_, and *χ*_*4*_ indicate proportion of the cells with corresponding genotype.

To smooth it, we optimize:minK,nF(K,n;MCFdel)=[∫00.5(K(MAF)n[1+K(MAF)n]−2×MAF+MAF×MCFdel)2dMAF].

Assume K^ and n^ are optimal solution given MCFdel, for mutations of *TP53* and *BIRC3*,MCF=K^(MAF)n^[1+K^(MAF)n^].

The above methods have been applied to adjust the data reported in this study before the TEDG analysis. New analysis based on adjusted data shows that: 1) all results of the TEDG analysis are totally identical to the previous version; 2) the growth rate of the late events is still significantly higher than that of early events (p-value=0.0259). This adjustment not only reduces the potential bias caused by copy number alterations, but also addresses another concern mentioned in Minor comment #4 about the integration of FISH data and NGS data.

To modify our manuscript:

A) We have updated all relevant data in this manuscript, updated the description of TEDG framework in Figure 1, and added the following sentence to the legend of the Figure 1:

“Then we use mutation cell frequency (MCF) to adjust and unify the data (middle panel).”

B) We have added the following sentence in Results, “Tumor Evolutionary Directed Graphs”:

“Specifically, the techniques of high-depth next generation sequencing (NGS) and fluorescence in situ hybridization (FISH) are separately carried out to assess the mutation allele frequency (MAF) and copy number abnormality (CNA) of selected driver genes. To unify both types of data, and to adjust the MAF of mutations in genes with CNA, we introduce mutation cell frequency (MCF) for quantification of genetic lesions (Materials and methods, Figure 1—figure supplement 1).”

C) We have put Figures 7, 8, 9 and 10 in Figure 1—figure supplement 1, and added expanded the Materials and methods accordingly.

*1) Relative timing of mutations: it is not entirely clear in the Materials and methods or main text exactly how the temporal order of mutations is inferred. At one point, mutations are described as 'present' or 'absent' on the basis of allele fraction >5% and compared across time points. Really, the only true method for inferring temporal ordering is through reconstruction of phylogenetic trees; this can be difficult from bulk sequencing data, but there are methods emerging to do this*.Author response image 5.Relative timing of mutations of 70 patients.Each column represents one patient with at least two time points. Magenta (MCF>5%, present) and blue bars (absent), which are ordered by time information, indicate the mutation status of the corresponding alteration. For one patient, if the present of alteration A is earlier than B, we claim A predates B.

In our work, the relative timing of mutations is inferred by longitudinal data with multiple time points. If the presence of mutation A (allele fraction >5%) predates that of mutation B, mutation A is considered earlier than mutation B (Figure 11).Author response image 6.Phylogenetic trees of CLL patients.Twenty-one out of all CLL patients contain the change of mutation status during disease progression. The phylogenetic trees are constructed based on mutation status of the driver genes. Green balls are normal cells, while all the others are cancer cells with particular alterations.

To explain this method clearly in the manuscript, we put Figure 11 as Figure 1—figure supplement 2, and add the following sentence in Results, “Tumor Evolutionary Directed Graphs”:

“Particularly, we include CNA represented in >5% of leukemic cells by FISH and mutations with MCF>5% (see examples of CLL patients in Figure 1—figure supplement 2). First, if a given genetic lesion is observed to be temporally earlier than another lesion, we connected them with a directed edge to represent their sequential order of development (Figure 1).”

We agree that the construction of phylogenetic trees is a good option to infer temporal orders of gene mutations by assuming common ancestors are early events, but limited to the fact that only very few genes have been considered in this study, it is difficult to reconstruct an accurate phylogenetic tree of patients. To approximate the phylogenetic trees, we define the pairwise distances between samples by considering both the number of common mutations and the number of different mutations, and make use of the complete linkage distance method to construct the phylogenetic trees of wild type cell, tumor cell at diagnosis, and tumor cell at different stages of relapse (Figure 12). Other methods, such as neighbor-joining and parsimony, produce similar results, as expected from the small number of branches and informative sites (driver mutations that vary within a tumor) and, as described below, branching assessment using bootstrap is generically non-significant.Author response image 7.Change of mutation frequency of patient #1.There are four samples for patient #1. Mutation frequencies of different alterations change according to the progression of the disease.

In most of cases, phylogenetic tree method indicates the same order of mutations as our longitudinal method used in TEDG. However, the order of gene mutations inferred by longitudinal method and phylogenetic tree method are not totally identical. For example, the phylogenetic tree of patient #1 (Figures 12 and 13) indicates del13q and delRB1 are early events, and then two branches are independently developed with one harbors *SF3B1*(K700E) and del11q, and the other harbors *SF3B1*(K666E). The first branch then develops *SF3B1*(K666E) and delBIRC3. The phylogenetic tree method does not predict the order of delBIRC3 and the mutation of *SF3B1*(K666E), but according to Figures 11 and 13, in patient #1, mutation of *SF3B1*(K666E) predates delBIRC3. The temporal order of delBIRC3 and mutation *SF3B1*(K666E) may not occur in the same subclone, but it indicates the competition between two subclones, suggesting subclone with delBIRC3 may have stronger fitness in this particular patient. Therefore, different from the phylogenetic tree method, the longitudinal analysis of TEDG presents the temporal orders of mutations, not only the results of the disease progression within a subclone but also consequences of competition among different clones.

To compare the results between the phylogenetic trees method and TEDG, we have applied both methods in our CLL data. The results show that all significantly early or late events with p-value<0.01 (**) reported by TEDG (before or after adjustment of MAF) are still significant (Figure 14). *NOTCH1*, reported significant early in TEDG, is very early (not significant due to limited number of samples). Furthermore, we rank all events based on fold change between indegree and outdegree by phylogenetic tree method and TEDG separately. Strikingly, the rank of all events based on those two methods is significantly related with Pearson correlation more than 0.9 (Figure 14). In the TEDG network analysis, the backbone of CLL evolution is slightly changed, but two main branches are the same, indicating two major types of CLL patients suffering T12 and del13q independently (Figure 14). The actual topologies of the phylogeny inferred network and TEDG are very similar (six out of ten edges are in common and share directionality, p-value<0.0001 by Fisher’s exact test). It is interesting to observe that the differences are in the order of del17p and *TP53* mutations and the indirect association of *SF3B1*.Author response image 8.The comparison between TEDG (B) and phylogenetic tree method (A).The two methods are compared in terms of the order of mutations and the TEDG networks. * indicates p-value<0.05, and ** indicates p-value<0.01. p-value in (C) is calculated by Fisher’s exact test.

We would like also to emphasize that phylogenetic methods usually require higher number of alterations than the ones considered in our manuscript. Standard statistical techniques to assess the branching for each tree rely on bootstrapping segregating sites. The statistical power of these techniques using longitudinal data from cancer patients in few selected driver genes is extremely limited. None of the standard phylogenetic methods described above produce branches with enough bootstrap support (>80%). On the other hand, TEDG works with aggregate data from several patients allowing statistical power for robust estimates.

In summary, we introduce a phylogenetic tree method to compare with our longitudinal method, and show that although the two methods are slightly different in capturing the effect of competition of different subclones, the main results inferred from both methods are almost the same, indicating the robustness of our overall analysis.

To modify our manuscript, we add a new paragraph in the Discussion section:

“Phylogenetic trees are often employed to infer temporal order of gene mutations by assuming that common ancestors are early events, but phylogenetic methods usually require higher number of alterations than the ones available in our cohort. Standard statistical techniques to assess the significance of different branches rely on bootstrapping segregating sites, and the statistical power of these techniques using longitudinal data from cancer patients in few selected driver genes is extremely limited. None of standard phylogenetic methods such as, distance matrix methods, Bayesian phylogenetic methods or parsimony methods, can produce branches with enough bootstrap support (>80%). On the other hand, TEDG works with aggregate data from several patients allowing statistical power for robust estimates.”

*3) Rate of increase: The argument that later mutations show more rapid increase than earlier mutations is difficult to sustain on the basis of these data. The initiating lesion, by definition, will be present in all the tumor cells, and therefore cannot increase at the same magnitude as later mutations. Given that it is a prerequisite for entry into the study that the patient be diagnosed with CLL, one would imagine that at that diagnostic time point, all the earliest mutations might already be clonal, and therefore will not change over time*.

To address this concern, we only consider the growth rate of subclonal mutations. Particularly, clonal mutations with MCF greater than 20% are eliminated from the comparison. As shown in Figure 15, late events are significantly faster than early events with p-value =0.0483 (Wilcoxon rank-sum test), indicating late events grow fast is not only because of the limitations on early events.

To clarify this point in the manuscript, we have added Figure 15 as Figure 6—figure supplement 3, and the following sentence in Results, “Rate of Allele Frequency Change”:

“Note that the initiating lesions are usually clonal, presenting in most of the tumor cells, and therefore do not increase in frequency at the same magnitude as subclonal mutations. However, eliminating clonal genetic lesions with MCF>20% in the above analysis, we still find the growth rates of late and early events to be significantly different with p-value=0.0483 (Figure 6—figure supplement 3).Author response image 9.Comparison of the growth rate of subclonal alterations between early and late events.

*4) A question for another type of inference approach is: how do we know ground truth? The only way to know is if there were a set of samples from which there was frequent sampling, and if there were within clinical defined groups. Robustness of the network is dependent on how good the data is going in. While they do mention that they selected 70 samples for TEDG for which there were at least two sequential samples, in all likelihood, 2 samples are inadequate for defining the hierarchical relationships. There needs to be some table that provides information on the distribution of sequential samples per patient, and what types of time intervals between sequential samples. While a figure about the samples is provided as*
Figure 3—figure supplement 1*, this is very hard to digest. This is a very heterogeneous dataset, with 10 of 70 proceeding to Richters, more than half with treatment. A question is whether or not there are enough samples, by the time that one breaks down the patients into discrete clinical groups, to understand the hierarchical relationships. Related to this, the numbers of samples are very confusing. This is what the reviewer understood: it seems that they start out with 202 patients, but only 70 had known highly recurrent drivers. 10 of the 70 had Richters Transformation, and how many have been excluded? By the time we get to*
Figure 4*, there are 32 samples, but there is no discussion how this number came about. In general, in what percentage of sample was a polyclonal tumor population or a subclonal driver identified at the time of diagnosis?*

We do not know the ground truth, and this is the main purpose of our and other methods to infer the order of alterations from longitudinal data. However, we have performed simulations where we start with the ground truth, a model of how mutations accumulate and a sampling strategy. In particular, we tested the TEDG method in an artificial example by separately simulating linear evolution and branching evolution of cancer as one-step Markov process. The simulation shows that TEDG’s accuracy in the linear model reaches 80% when N is 30 (Figure 2). Similarly, the accuracy reaches 90% when N is 30 in branching model (Figure 2, Materials and methods). Simulations allow estimating the power of our data and method to assess the order of alterations and the different evolutionary trajectories.Author response image 10.Summary of longitudinal data in 70 patients.(A) The 70 patients are selected from a big cohort of 1,403 CLL patients with no-bias screening. (B) 70 patients are ranked according to their minimal cell frequency at diagnosis. Patient with minimal cell frequency less than 20 are in red, the others are in green.

Regarding the number of samples considered in each analysis, we are sorry for the confusion of the presentation of sample size in Figure 3—figure supplement 1 and Figure 4. To clarify the samples used in our study, we present our data in Figure 16. No biases are present in the case series we have utilized in the manuscript. Indeed, we have collected 14,03 newly diagnosed CLL patients between the year of 2001 and the year of 2012, during which 202 patients contain samples of multiple time pints. We systematically genotyped all the sequential samples collected at each progression (for progressive cases) and at the last follow-up (for non- progressive cases) from consecutive cases recruited in our database. Out of those 202 patients, 70 are informative for *TP53*, *NOTCH1*, *SF3B1*, *MYD88* or *BIRC3* mutations (which are the most common mutations in this leukemia), overall accounting for 164 paired sequential samples collected at diagnosis, progression and last follow up (Figure 16). Our study is mainly focusing on those 70 patients. The mutation information, as well as the number of sequential samples of each patient is shown in Figure 11. In total, we have 50 patients with two time points, 16 with three time points, and 4 with four time points. In Figure 4, we did not say there are 32 samples. We used 70 CLL patients to analyze evolution network of 32 alterations (one gene can have more alterations by considering mutation sites).

To answer the question that “*in what percentage of sample was a polyclonal tumor population or a subclonal driver identified at the time of diagnosis*”, we have added the distribution of minimal cell frequency at diagnosis as shown in Figure 16. Half (35 out of 70) of the patients have at least one subclonal mutation with cell frequency less than 20% at diagnosis.

To modify the manuscript:

A) We have added the following sentence in Results, “Tumor Evolutionary Directed Graphs”:

“To test TEDG method and also to show how many patients are required to approximate the ground truth, we employ the TEDG method in an artificial example by both simulating linear evolution and branching evolution of cancer, where the longitudinal data is generated by one-step Markov process and Nordling’s multi-mutation model.”

B) We have included Figure 16 to replace Figure 3—figure supplement 1, and added the following sentence in Results, “Tumor Evolutionary Directed Graph of CLL”:

“Half (35 out of 70) of the patients have at least one subclonal genetic lesion with cell frequency less than 20% at diagnosis (Figure 3—figure supplement 1).”

[Editors' note: further revisions were requested prior to acceptance, as described below.]

We have addressed the remaining concerns about the justification of defining Mutation Cell Fraction (MCF) and the application to FISH analysis. Following the suggestions: we have 1) further clarified the definition of MCF in manuscript, 2) described the variations of MAF (VAF) caused by sampling error based on dilution experiments, 3) included FACS analysis to consider tumor purity to calibrate MCF, 4) justified the Hill function form of MCF by comparing with more simplistic definitions, and 5) compared the calibrated MCF with previously defined Cancer Cell Fraction (CCF) by ABSOLUTE 1.0.6 (26).

*1) A major revision has been the addition of the concept of MCF. This was done to address the concern of adjusting for local copy number changes that could affect estimation of the clone size with a gene alteration. Unfortunately, the approach taken by the authors to calculate MCF seems not well defined. This reviewer is concerned that sampling error in VAF values might fluctuate above or below 0.5 in ways that have a large influence on the inferred MCF. No convincing data is presented by the authors to justify their novel and elaborate seeming approach*.

*Several approaches for correcting mutation VAF values for tumor purity and somatic copy number changes have been described (e.g.*
*Carter et al., 2012**, Nat Biotech;*
[26]*, Cell;*
*Fischer et al., 2014**, Cell Reports). Notably, these methods rely on analysis of germline heterozygous SNPs at the mutant locus to infer CN-LOH, which is a much more robust approach. Could the authors simply use one of these established (and principled) approaches? This would leverage their exome data more fully*.

1) To clarify the definition of MCF, we have revised manuscript as follows:

Excerpt from Results, “Tumor Evolutionary Directed Graphs”:

“To unify both types of data, and to adjust the MAF of mutations in genes with CNA, we introduce mutation cell frequency (MCF, defined as the fraction of tumor cells with a particular alteration) for quantification of genetic lesions (Materials and methods, Figure 1—figure supplement 1).”

Excerpt from Materials and methods, “Adjustment of Mutation Frequency”:

“For instance, if tumor purity is 70% and MAF of a heterogeneous variance in a diploid region is 0.2, then MCF is approximated by 0.2×2/0.7≈0.57.”

2) To estimate the variation of MAF (VAF) caused by sampling error, we have employed a bioinformatic approach to call mutations of low abundance out of the background error noise of deep sequencing data. Based on dilution experiments, we calibrated for systematic biases that lead to sequencing errors and derived the depth distribution of sequencing errors to be negative binomial. We determined that the sequencing depth of 1,000x is sufficient for a highly sensitive detection of mutations with allele frequency >1% out of the background error noise and variance of the estimate is small when allele frequency is larger than the detection limit. A detailed account of calibration of estimates using ultra-deep sequencing has been described in a related publication (Figure 3 of [42]*,* referenced in the manuscript).Author response image 11.The impact of sampling error on MCF.Confidence interval of MCF is inferred based on the variation of MAF estimated by dilution experiments.

To show how the sampling error affects MCF, we calculate the 95% confidence interval of MCF based on variation of MAF described by [42]. Particularly, the 95% confidence interval of MAF at 0.5%, 2.5%, 5%, and 25% are used to estimate upper and lower bound of MCF with the formula described in Materials and methods, “Adjustment of Mutation Allele Frequency”, and Figure 1—figure supplement 1. The length of 95% confidence interval of MCF are respectively 0.1%, 0.6%, 1.2%, and 2%, indicating that the sampling error slightly fluctuates the value of MCF; however, only 0.2% (1 out of 415) of the detected mutations, which is on the boundary of our cutoff, will be affected by this fluctuation, not changing the overall results (Figure 17).

To modify the manuscript, we have added the following sentence at the end of the Materials and methods section, “Estimation of the Variant Allele Frequency”:

“The sequencing depth in this study is on average 1200x, which is sufficient for a highly sensitive detection of mutations with allele frequency >1% out of the background error noise (42).”

3) Our estimation of MCF in the first revision of manuscript was based on two assumptions: 1) there are no copy neutral LOHs (cn-LOH) and 2) tumor purity is very high. The two assumptions may limit the application of this approach in cn-LOH regions or low-purity samples.

The reviewers have suggested considering neighboring SNPs to address the concern of cn-LOH. Unfortunately, we do not have whole-exome or SNP array data for most of our longitudinal cases. However, whole-exome sequencing data are available for 10 out of 70 patients, including (a) four patients with paired tumor and normal control (published recently in Messina et al., Blood, 2014 and discussed below) and (b) six patients with only tumor samples (published recently in [12], JEM, and discussed below) in our cohort. To check whether the driver genes (i.e., *TP53*, *SF3B1*, *BIRC3*, *MYD88*, *NOTCH1)* are affected by cn-LOH in those ten patients, we have checked the neighboring SNPs in IGV viewer, and find none of our driver genes in the regions of cn-LOH. In a recent study, Pfeifer et al. have analyzed 70 CLL patients, and listed the regions of cn-LOH in Table 4 of their paper (Pfeifer et al, 2007, Blood). We compare all our diver genes with their results, and find none of our genes in their list. In summary, based on limited cases with heterozygosity information in the list of driver genes selected in these studies and in published results by an independent group in an extended cohort, we reason that cn-LOH is not a major effect in the targeted genes of our current study.

To consider tumor purity, we have included FACS analysis of CD19/CD5 expression in the revised manuscript. CD19+CD5+ markers represent a gold standard to define the representation of CLL cells in a diagnostic or research sample, as these tumors characteristically and invariably express these molecules on their surface, in contrast with normal peripheral B cells.

To modify the manuscript, we have added the following sentences in Materials and methods, “Adjustment of Mutation Frequency”:

“To consider tumor content we performed CD19/CD5 FACS analysis to quantify the fraction of tumor cells.”

“The MCF values were then divided by the fraction of CD19+CD5+ cells based on FACS.”

We have also added the following new section in Materials and methods:

“Fluorescence-activated cell sorting (FACS) analysis

The count of CD19+CD5+ cells is a standard assay to define the representation of CLL cells in a diagnostic or research sample as measured by Fluorescence-activated cell sorting (FACS) analysis. A FACSCalibur flow cytometer (Becton-Dickinson) was utilized for the analysis. Expression of CD5 and CD19 was analyzed by combining Peridinin-Chlorophyll-Protein-Cyanine-5.5 (PerCP-Cy5.5)-conjugated anti-CD19 mAbs and fluorescein isothyocyanate-conjugated anti-CD5 mAbs. In order to estimate the proportion of cells co-expressing CD19 and CD5, for each sample events were acquired by gating on low forward and side scatter (FSC/SSC) CD19+ cells, which were further divided into CD5− and CD5+ subsets. Irrelevant isotype-matched antibodies (Becton-Dickinson) were used to determine background fluorescence. FACS data of all samples are listed in [Supplementary-material SD2-data].”Author response image 12.Justification of MCF.X-axis indicates fraction of CD19+CD5+ cells assessed by FACS analysis, and y-axis indicates maximal mutation fraction of all targeted driver genes of each sample calculated by different methods. One blue dot represents one sample, and contours indicate the density of dots. A suitable calculation of maximal driver mutation fraction will approximate but not exceed the fraction of cancer nuclei. The upper red line indicates CD19+CD5+ cell fraction, and the lower red line indicates a 20% lower interval of it. Apparently, tumor purities of 55 samples are properly assessed by the Hill function MCF, which is better than both MAF without adjustment (10 samples) and simple piecewise MCF (27 samples).

4) To further justify the Hill function form of MCF, we have used data from FACS analysis to compare it to more simplistic methods. Particularly, given one sample, the maximal mutation fraction of all drivers is separately calculated based on three approaches: MAF, piecewise MCF, and Hill function MCF. A suitable calculation of this score will approximate but not exceed the fraction of cancer nuclei estimated by FACS analysis of the fraction of CD19+CD5+ cells. The comparison shows that the Hill function form of MCF is more robust to assess the tumor purity (Figure 18).Author response image 13.Comparison of MCF and CCF.The samples with WES available show that our MCF falls within the estimates of ABSOLUTE. ABSOLUTE underestimates tumor purity in sample #37 as compared to CD19/CD5 FACS and our MCF estimate.

To modify the manuscript, we have used Figure 18 as Figure 1—figure supplement 1, and added the following sentence in Materials and methods, “Adjustment of Mutation Frequency”:

“Additionally, by comparing to FACS measurement of tumor purity, this Hill function form of MCF is more robust than either MAF without adjustment or simplistic piecewise function in assessing the fraction of cancer nuclei (Figure 1—figure supplement 1).”

5) As mentioned by the reviewers, several approaches for correcting MAF have been described recently. ABSOLUTE (Carter et al., 2012, Nat Biotech) infers tumor purity and average cancer cell ploidy from the analysis of SNP array and somatic DNA alterations. ABSOLUTE 1.0.5 ([26], Cell) and 1.0.6 were further released to include the calculation of cancer cell fraction (CCF) for each sSNV. More recently, cloneHD (Fischer et al., 2014 Cell Reports) was published to reconstruct the subclonal structure of a population from sequencing data. All these approaches have been proved to be robust in the analysis of cancer genome, but all methods require genome-wide information, such as SNP array, whole-exome or whole-genome sequencing, which are only available for a few samples in our analysis. We want to emphasize that compared to the other approaches this manuscript presents the analysis of a few targeted alterations in large number of longitudinal samples (70 longitudinal patients).Author response image 14.ABSOLUTE report of case #37.Several models fit the data with different purity estimates.

To compare our method to genome-wide methods, we have analyzed previously published whole-exome sequencing (WES) and SNP array data of four of our patients: #57, #4, #37, and #59 ([12], JEM, and Messina et al., 2014, Blood. SNP array data for the other six patients are not available). We ran ABSOLUTE 1.0.6 with the parameters min.ploidy=1, and max.ploidy=3 (given that all samples in the research of Landau et al. ([26], Cell) were estimated to have near-diploid DNA content). CCFs (cancer cell fraction as defined by ABSOLUTE) are calculated. By comparing MCF with the confidence interval of CCF, for mutations of 3 out of 4 cases, the results are consistent (Figure 19). In patient #37, however, according to ABSOLUTE, MCF of a *TP53* mutation (MAF=0.34) is close to 100% because the tumor content is estimated to be ∼40%, which is not consistent with our FACS analysis (CD19+CD5+ cell fraction is 95%). This indicates that in this discrepant case, ABSOLUTE underestimates the tumor purity (40%) compared to FACS (95%). We were extremely curious about this underestimation and upon further investigation we find that ABSOLUTE reports different models that fit the data, and in fact there are several solutions that coincide with our estimate of MCF and the FACS analysis (Figure 20).

*2) They rely on FISH analysis for estimations of trisomy 12 and deletions of various chromosomal regions. This is not accurate, as it is not possible to correct for purity when using FISH data. In addition, copy neutral LOH cannot be inferred from FISH*.

To address this concern, we have included FACS analysis. We performed the analysis with corrected tumor content based on CD19/CD5 FACS analysis and all the major results are the same as expected from the high tumor content of these samples (average 83%). Specifically, we updated all MCFs dividing by the fraction of CD19+CD5+ cells. After this correction, MCFs of a few number alterations, which are not considered before, exceed the cutoff of 5%, including del13q (delRB1) at the first time point of patients #13 and #58, and *NOTCH1* mutation at the second time point of patient #34. Accordingly, Figure 1—figure supplement 2, Figure 3 were updated (edges in the network plot, indegree, outdegree, etc.). Importantly, this modification did not change TEDG (Figure 3) or the order of mutations (Figure 3), indicating our analysis is robust to this adjustment.

As described earlier in response to Major comment 1, we have found that cn-LOH is not a major effect in the targeted genes of our current study.

To modify the manuscript, we have updated this data to all analysis in our manuscript, including the MCFs in Table 3, and modifications in figures: change of some arrows in Figure 4, indegree/outdegree in Figure 3—figure supplement 2 and Figure 4—figure supplement 1, boxplots in Figure 6 and Figure 6—figure supplement 3, etc.

Also we have added the following sentence in Materials and methods, “Adjustment of Mutation Frequency”:

“MCF of a copy number abnormality (i.e. T12, del11q, del13q_x1, del13q_x2, del17p, delBIRC3, delRB1_x1, delRB1_x2) was calculated by FISH analysis divided by the fraction of CD19/CD5 cells based on FACS, correcting for the tumor purity.”

Additionally, we have also added a supplementary table with detailed data of CD19/CD5 as [Supplementary-material SD2-data].